# Molecular mechanism for the synchronized electrostatic coacervation and co-aggregation of alpha-synuclein and tau

Pablo Gracia[1,2,6], David Polanco [1,2,6], Jorge Tarancón-Díez [1,2], Ilenia Serra [3], Maruan Bracci[3], Javier Oroz [4], Douglas V. Laurents [4], Inés García [3,5] & Nunilo Cremades [1,2] ✉

Amyloid aggregation of α-synuclein (αS) is the hallmark of Parkinson's disease and other synucleinopathies. Recently, Tau protein, generally associated with Alzheimer's disease, has been linked to αS pathology and observed to co-localize in αS-rich disease inclusions, although the molecular mechanisms for the co-aggregation of both proteins remain elusive. We report here that αS phase-separates into liquid condensates by electrostatic complex coacervation with positively charged polypeptides such as Tau. Condensates undergo either fast gelation or coalescence followed by slow amyloid aggregation depending on the affinity of αS for the poly-cation and the rate of valence exhaustion of the condensate network. By combining a set of advanced biophysical techniques, we have been able to characterize αS/Tau liquid-liquid phase separation and identified key factors that lead to the formation of hetero-aggregates containing both proteins in the interior of the liquid protein condensates.

Besides membranous compartments, spatial segregation in the cell can be achieved through the formation of protein-rich liquid-like dense bodies, termed biomolecular condensates or liquid droplets, through a process known as liquid-liquid phase separation (LLPS)[1]. These liquid droplets are formed by multivalent transient interactions, commonly between proteins or proteins and RNA, and serve a myriad of functions in essentially all living systems[2]. A large number of proteins that can undergo LLPS present low-complexity sequences that remain largely disordered in native conditions, as well as when forming biomolecular condensates[3–5]. A number of experimental studies have highlighted the flexible, typically disordered and multivalent nature of the constituent proteins in these liquid-like condensates[6], although less is known about the particular molecular determinants governing the growth and maturation of these condensates into more solid-like states.

Emerging evidence supports the hypothesis that aberrant protein-driven LLPS and the transition of the liquid droplets to solid-like structures might be a relevant cellular pathway leading to the formation of insoluble, toxic aggregates that are often a hallmark of degenerative diseases. A number of LLPS-associated intrinsically disordered proteins (IDPs), often highly charged and flexible, have been long linked to neurodegeneration through amyloid aggregation processes. In particular, biomolecular condensates of IDPs such as FUS[7] or TDP-43[8], or proteins with large low-complexity domains such as hnRNPA1[9], have been shown to age into gel-like or even solid-like structures, through a process known as liquid-to-solid phase transition (LSPT), as a function of time or as a response to certain post-translational modifications or pathology-related mutations[1,7].

Another IDP that has been related to in vivo LLPS is Tau, a microtubule-associated disordered protein whose amyloid aggregation

[1]Institute for Biocomputation and Physics of Complex Systems (BIFI), University of Zaragoza, 50018 Zaragoza, Spain. [2]Department of Biochemistry and Molecular and Cell Biology, University of Zaragoza, 50009 Zaragoza, Spain. [3]Department of Condensed Matter Physics, Faculty of Sciences, University of Zaragoza, 50009 Zaragoza, Spain. [4]"Rocasolano" Institute for Physical Chemistry, CSIC, Serrano 119, Madrid E-28006, Spain. [5]Centro Universitario de la Defensa, Academia General Militar, Ctra. de Huesca s/n, 50090 Zaragoza, Spain. [6]These authors contributed equally: Pablo Gracia, David Polanco. ✉e-mail: ncc@unizar.es

is associated with Alzheimer's disease[10], but also more recently linked to Parkinson's disease (PD) and other synucleinopathies[11–13]. Tau has been shown to de-mix from the solution/cytoplasm as a consequence of favorable electrostatic interactions[14], resulting in liquid Tau-rich droplets which are referred to as electrostatic coacervates. Such type of non-specific interactions have likewise been observed to be the driving force of numerous biomolecular condensates in nature[15]. In the case of Tau, electrostatic coacervates can be formed through either simple coacervation, where oppositely charged regions of the protein trigger the de-mixing process[14], or complex coacervation through the interaction with negatively charged polymers such as RNA[16].

Recently, α-synuclein (αS), an amyloidogenic IDP involved in PD and other neurodegenerative disorders, collectively referred to as synucleinopathies[17,18], has been shown to concentrate in protein condensates with liquid-like behavior both in cellular and animal models[19,20]. In vitro studies have proposed that αS undergoes LLPS by simple coacervation through primarily hydrophobic interactions, although this process requires particularly high protein concentrations and atypical long incubation times[19,21]. Whether the αS-containing condensates observed in vivo are formed through this or other LLPS processes is a key question that remains unsolved. Similarly, although αS amyloid aggregates are observed inside neurons in PD and other synucleinopathies, the precise mechanisms by which αS undergoes amyloid aggregation inside cells are still unclear, as the sole overexpression of the protein seem to be unable to trigger this process. Additional cellular insults are typically required[22], suggesting that particular cellular locations or microenvironments are necessary for the de novo nucleation of αS amyloid assemblies inside cells. A particularly aggregation-prone cellular environment could be the interior of protein condensates[23].

Intriguingly, αS and Tau have been found to co-localize in the disease hallmark inclusions of individuals with PD and other synucleinopathies[24,25] and a synergistic pathological relationship between these two proteins has been experimentally reported[26,27], suggesting a potential cross-talk between αS and Tau aggregation in neurodegenerative disorders. αS and Tau have been found to interact and promote each other's aggregation both in vitro and in vivo[28,29], and hetero-aggregates composed of both proteins have been observed in the brains of patients suffering from synucleinopathies[30]. However, the molecular basis underlying the interplay between αS and Tau, and the mechanisms of their co-aggregation are poorly understood. αS has been reported to interact with Tau through electrostatic attractions between the highly negatively charged C-terminal region of αS and the central proline-rich region of Tau, which is also enriched in positively charged residues[31].

In this study, we show that αS can indeed phase-separate into liquid droplets in the presence of Tau by electrostatic complex coacervation, similarly as it does with other positively charged polypeptides like poly-L-lysine (pLK) and that, in this process, αS acts as scaffolding molecule of the droplet network. We have identified dramatic differences in the maturation processes of the αS electrostatic coacervates, which are related to the differences in the valences and interaction strengths of the proteins involved in the coacervate network. Interestingly, we have observed amyloid co-aggregation of αS and Tau inside the long-lived liquid coacervates and have determined some of the key factors that lead to co-aggregation of both proteins inside this type of coacervates. This process, which we have characterized here in detail, constitutes a possible molecular mechanism underlying the co-localization of both proteins in disease-hallmark inclusions.

## Results

### αS forms electrostatic complex coacervates with poly-L-lysine
αS has a highly anionic C-terminal tail at neutral pH (Fig. 1a), which we hypothesized could undergo LLPS by electrostatic complex coacervation with poly-cationic disordered polypeptide molecules. As an initial model molecule, we used poly-L-lysine of 100 residues (pLK), given the positively charged and disordered polymeric nature of this molecule at neutral pH[32]. First, we confirmed that pLK interacts with the $C_t$-domain of αS by solution NMR spectroscopy (Fig. 1b) using $^{13}C$/$^{15}N$-labeled αS in the presence of increasing αS:pLK molar ratios. The interaction of pLK with the $C_t$-domain of αS is evident by both chemical shift perturbations and the decrease in peak intensities in this protein region. Interestingly, when we mixed αS with pLK at αS concentrations of ca. 5–25 μM in the presence of polyethylene glycol (5–15% PEG-8) (the typical LLPS buffer: 10 mM HEPES pH 7.4, 100 mM NaCl, 15% PEG-8), we immediately observed protein droplet formation by widefield fluorescence (WF) and brightfield (BF) microscopy (Fig. 1c). Droplets of 1–5 μm in size, containing condensed αS (spiked with 1 μM AlexaFluor488-labeled αS, AF488-αS) are readily formed, and their electrostatic nature is evident from their resistance against 10% 1,6-hexanediol (1,6-HD) and their sensitivity to increasing NaCl concentrations (Fig. 1c). The liquid-like nature of the αS/pLK electrostatic complex coacervates was proved by their ability to fuse within milliseconds (Fig. 1d). By means of turbidimetry, we quantified droplet formation under these conditions, confirmed the electrostatic nature of the main interactions involved in their stability (Fig. 1e), and evaluated the effect of different polymer ratios on the LLPS process (Fig. 1f). Although droplet formation is observed within a wide range of polymer ratios, the process is highly favored when pLK is in excess with respect to αS. LLPS was also observed when a chemically different crowding agent, dextran-70 (70 kDa), was used or when different sample formats are employed (including drop-on-glass slide, microplate well of different materials, or Eppendorf or quartz capillary tubes).

### αS forms electrostatic complex coacervates with Tau by liquid–liquid phase co-separation
Based on our observation of αS/pLK electrostatic complex coacervation, and the previous observation of αS as a client molecule of Tau/RNA condensates[31] through a direct interaction with Tau, we hypothesized that αS and Tau could co-separate from the solvent in the absence of RNA by electrostatic complex coacervation, being αS a scaffolding protein in the αS/Tau coacervates (see the Tau charge distribution in Fig. 2e). We observed that when 10 μM αS and 10 μM Tau441 (containing 1 μM AF488-αS and 1 μM Atto647N-Tau, respectively) are mixed together in LLPS buffer, they readily form protein condensates, which contain both proteins as seen by WF microscopy (Fig. 2a). The co-localization of both proteins in the droplets was confirmed by confocal (CF) microscopy (Supplementary Fig. 1a). A similar behavior was observed when dextran-70 was used as crowding agent (Supplementary Fig. 1c). By using FITC-labeled PEG or dextran, we found that both crowding agents distribute homogeneously over the whole sample without showing a segregative nor an associative behavior (Supplementary Fig. 1d). Instead, this indicates that they favor phase separation through macromolecular crowding effects in this system, being PEG a preferentially stabilizing crowding agent, as observed for other LLPS systems[33,34]. These protein-rich droplets are sensitive to NaCl (1 M) but not to 1,6-HD (10% v/v), thus confirming their electrostatic nature (Supplementary Fig. 2a, b). Observation of droplet fusion events within milliseconds by BF microscopy verify their liquid-like behavior (Fig. 2b).

In order to test the role of αS in this LLPS process, we first investigated the effect of αS on the stability of the droplets by turbidimetry using increasing NaCl concentrations (Fig. 2c). The higher values of light scattering (at 350 nm) with higher salt concentrations in the sample containing αS evidence a stabilizing role of αS in this LLPS system. A similar effect can be observed upon increasing the concentration of αS (thus the αS:Tau441 ratio) up to ca. 10-fold with respect to Tau concentration (5 μM) (Fig. 2d). To prove that αS is a

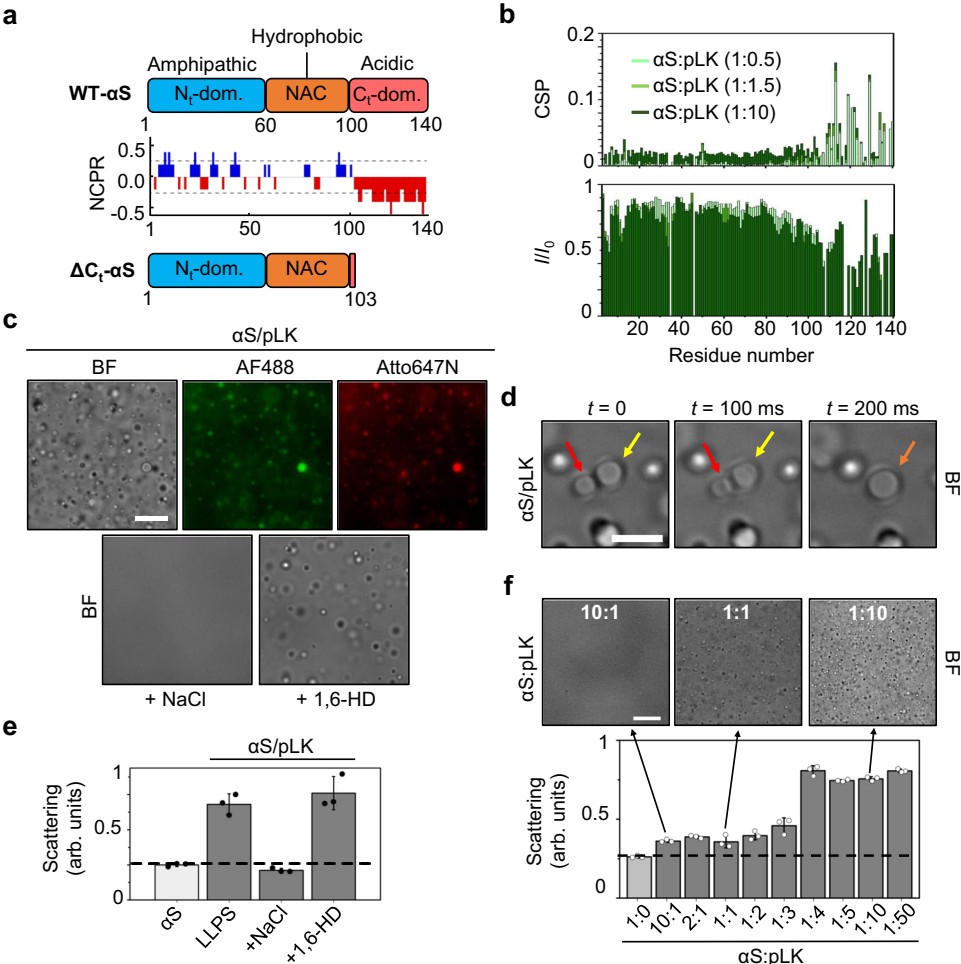

**Fig. 1 | Complex electrostatic coacervation of αS with poly-L-lysine. a** Schematic of the different protein regions in WT-αS and the ΔCt-αS variant used in this study. The amphipathic N-terminal domain, the hydrophobic amyloid-forming (NAC) region and the negatively charged C-terminal domain are shown in blue, orange and red, respectively. The net charge per residue (NCPR) diagram of WT-αS is shown. **b** NMR analysis of αS/pLK interaction in the absence of macromolecular crowding. Chemical shift perturbation (CSP, top) and intensity ratio ($I/I_0$, bottom) analysis of HSQC spectra of $C^{13}$, $N^{15}$-labeled αS (150 μM) in the presence of increasing concentrations of pLK (αS:pLK molar ratios of 1:0.5, 1:1.5 and 1:10 are shown in light green, green and dark green, respectively). **c** Brightfield (BF, top left and bottom images) and widefield fluorescence (WF, top center and right images) microscopy images of αS/pLK coacervates (1:10 molar ratio) at an αS concentration of 25 μM (1 μM AF488-labeled αS or Atto647N-labeled pLK for WF imaging) in LLPS buffer (top) or after adding 500 mM NaCl (bottom left) or 10% 1,6-hexanediol (1,6-HD; bottom right). Scale bar = 20 μm. **d** Representative BF microscopy images of an αS/pLK (1:10 molar ratio) droplet fusion at an αS concentration of 25 μM; arrows indicate individual liquid droplets (red and yellow arrows) merging into one new droplet (orange arrow) within 200 ms. Scale bar = 20 μm. **e** Light scattering (at 350 nm) of αS/pLK coacervates in LLPS buffer, at an αS concentration of 25 μM, before and after adding 500 mM NaCl or 10% 1,6-HD ($N$ = 3 sample replicas, mean value and SD are also indicated). **f** BF images (top) and light scattering analysis (at 350 nm, bottom) of αS/pLK coacervates at an αS concentration of 25 μM at increasing αS:pLK molar ratios ($N$ = 3 sample replicas, mean value and SD are also indicated). Scale bar = 10 μm. The scale bar in one image is indicative of the scale for all the images in the same panel. Source data are provided as a Source Data file.

scaffolding protein in the coacervates, we decided to investigate the behavior of a LLPS-impaired Tau mutant, lacking the negatively charged N-terminal region (residues 1–150, see Fig. 2e), termed ΔNt-Tau. We verified by WF microscopy and turbidimetry that ΔNt-Tau does not undergo LLPS by itself (Fig. 2f and Supplementary Fig. 2d), as it was previously reported[14]. However, when αS was added to the dispersed solutions of this truncated Tau variant, the LLPS process was fully restored with droplet densities resembling those of solutions of full-length Tau and αS under similar conditions and protein concentrations. This process can also be observed at low macromolecular crowding conditions (Supplementary Fig. 2c). The role of the C-terminal region of αS in the LLPS process was demonstrated by the inhibition of droplet formation when a C-terminus truncated variant of αS (ΔCt-αS), lacking residues 104–140 (Fig. 1a), was used instead of the full-length protein (Fig. 2f and Supplementary Fig. 2d). Co-localization of αS and ΔNt-Tau was confirmed by confocal fluorescence microscopy (Supplementary Fig. 1b).

In order to further validate the mechanism of LLPS between Tau441 and αS, an additional Tau variant was used; namely the paired helical filament core fragment (PHF) within the microtubule-binding domain (MTBD), also typically referred to as K18 fragment when containing the four characteristic repeat domains (see Fig. 2e). αS has been recently reported to bind Tau preferentially at the proline-rich domain that precedes the microtubule-binding domain in sequence[31]. However, the PHF region is also rich in positively charged residues (see Fig. 2e), particularly lysines (15% of the residues), which prompted us to test for the ability of this region to also contribute to αS/Tau complex coacervation. We observed that K18 is unable to trigger LLPS by itself (Fig. 2f) under the conditions tested (LLPS buffer with either 15% PEG or 20% dextran) at concentrations up to 100 μM. However, when we added 50 μM αS to 50 μM K18, rapid formation of protein droplets containing both K18 and αS was observed by turbidimetry (Supplementary Fig. 2d) and WF microscopy (Fig. 2f). As expected, ΔCt-αS was not able to rescue the LLPS behavior of K18 (Fig. 2f). We noted that for

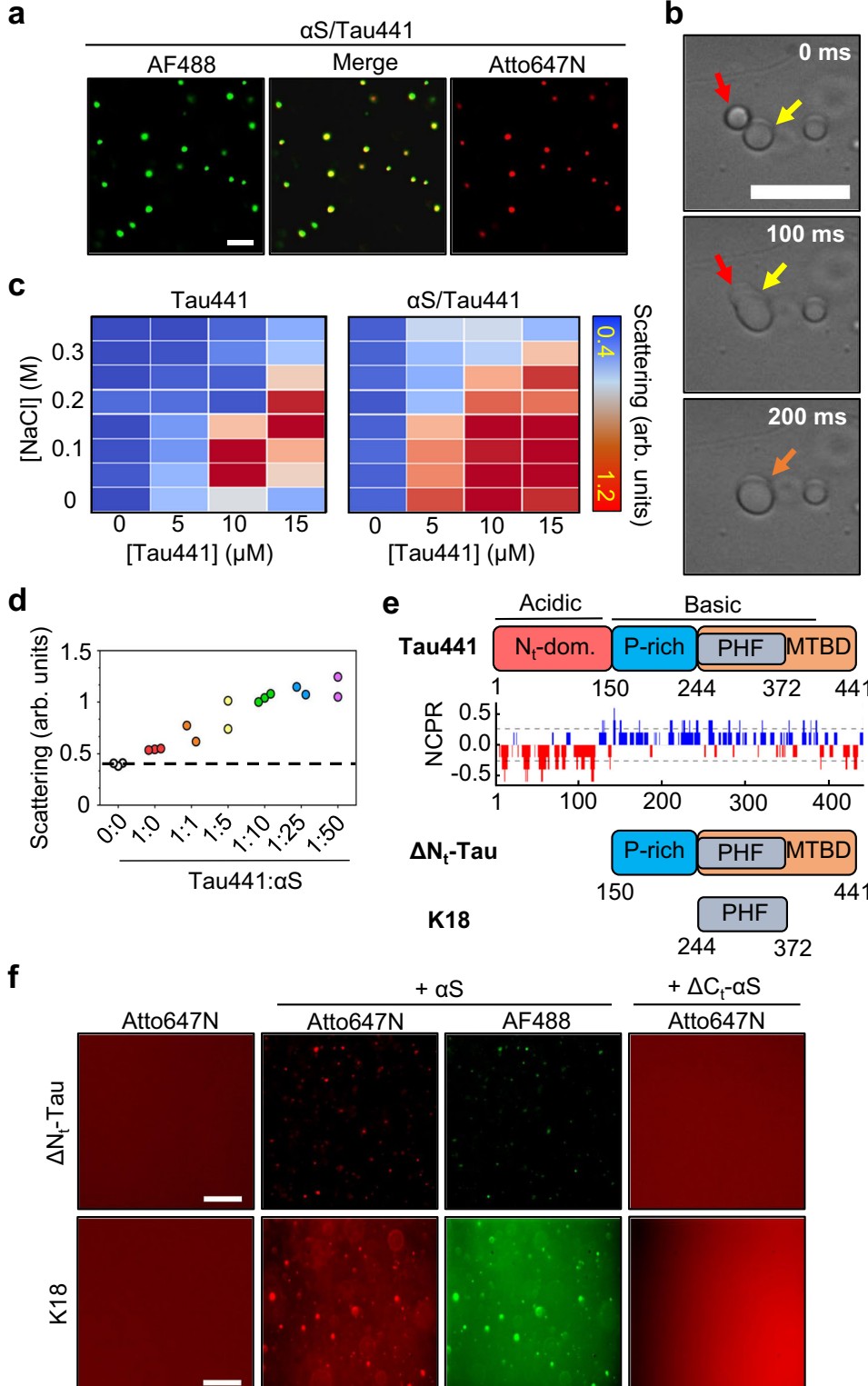

**Fig. 2 | Complex electrostatic coacervation of αS with Tau. a** Confocal (CF) microscopy images of αS/Tau441 coacervates in LLPS buffer (10 μM each protein, 0.5 μM AF488-labeled αS and Atto647N-labeled Tau441). **b** Representative differential interference contrast (DIC) microscopy image of an αS/Tau441 droplet fusion event (10 μM each protein). **c** Light scattering-based (at 350 nm) phase diagram of Tau441 LLPS (0–15 μM) in the absence (left) or the presence (right) of 50 μM αS. Warmer colors indicate more scattering. **d** Light scattering of αS/Tau441 LLPS samples with increasing concentrations of αS (Tau441 at 5 μM, $N = 2$–3 sample replicas, as indicated). **e** Schematic of some of the Tau protein variants used in this

study and the different protein regions: the negatively charged N-terminal domain (in red), the proline-rich region (in blue), the microtubule-binding domain (MTBD, in orange) and the amyloid-forming paired helical filament (PHF) region located within the MTBD (in gray). The net charge per residue (NCPR) diagram of Tau441 is shown. **f** WF microscopy images of αS or ΔCt-αS coacervation in LLPS buffer with ΔNt-Tau (top, 10 μM each protein) or K18 (bottom, 50 μM each protein), using 1 μM AF488-labeled αS and Atto647N-labeled ΔNt-Tau or K18. The scale bar in one image is indicative of the scale for all the images in the same panel (20 μm for panels **a**, **b** and **f**). Source data of panels **c** and **d** are provided as a Source Data file.

αS/K18 coacervation, slightly higher protein concentrations were needed to trigger LLPS as compared to αS/ΔNt-Tau or αS/Tau441 under otherwise identical conditions. This is consistent with a stronger interaction of the C-terminal region of αS with the proline-rich domain of Tau, as compared to the microtubule-binding domain, as previously observed[31].

Considering that ΔN$_t$-Tau is unable to undergo LLPS in the absence of αS, we chose this Tau variant as a model for αS/Tau LLPS characterization, taking into consideration that in the LLPS system with full-length Tau both simple (homotypic, Tau441/Tau441) and complex (heterotypic, αS/Tau441) coacervation processes occur simultaneously. We compared the extent of αS condensation (as fraction of the protein in the condensed phase, $f_{αS,c}$) in αS/Tau and αS/ΔN$_t$-Tau system by centrifugation and SDS-PAGE analysis of the disperse phase (see Supplementary Fig. 2e), and we found very similar values, being all the proteins at the same concentrations. More specifically, we obtained a $f_{αS,c}$ of 84 ± 2% and 79 ± 7% for αS/Tau and αS/ΔN$_t$-Tau, respectively, which indicates that heterotypic interactions between αS and Tau are preferred to the homotypic interactions between Tau molecules.

## Characterization of αS dynamics in electrostatic complex coacervates

The influence of the interactions with the different poly-cations and of the coacervation processes on the dynamics of αS was first investigated by means of fluorescence recovery after photobleaching (FRAP). We conducted FRAP assays (Fig. 3a–c) on αS/Tau441, αS/ΔN$_t$-Tau and αS/pLK coacervates (100 μM αS, supplemented with 2 μM αS AF488-αS, with either 100 μM Tau441 or ΔN$_t$-Tau, or 1 mM pLK). Data were acquired within the first 30 min after mixing the components of the sample. As can be seen from the representative FRAP images (Fig. 3a, αS/Tau441 coacervation) and their corresponding time-course curves (Fig. 3b, Supplementary Fig. 3), the dynamics of αS were very similar whithin the coacervates with Tau441 and ΔN$_t$-Tau, while significantly faster with pLK. The diffusion coefficient of αS inside the coacervates estimated from the FRAP data (as described by Kang et al.[35]) was $D = 0.013 ± 0.009$ μm$^2$/s and $D = 0.026 ± 0.008$ μm$^2$/s for αS/Tau441 and αS/ΔN$_t$-Tau, respectively, and $D = 0.18 ± 0.04$ μm$^2$/s for αS/pLK systems (Fig. 3c). The diffusion coefficient of αS in the dispersed phase was, however, orders of magnitude faster with respect to that of all the condensed phases, as determined by fluorescence correlation spectroscopy (FCS, see Supplementary Fig. 3) under identical conditions (LLPS buffer) but in the absence of poly-cation ($D = 8 ± 4$ μm$^2$/s). αS translational dynamics are, thus, greatly reduced within the coacervates as compared to the protein in the dispersed phase, due to a significant molecular crowding effect, although all the coacervates maintain a liquid-like nature within the first half an hour from their formation, with αS presenting faster dynamics within the condensates with pLK as compared to Tau.

Complementary to this, we studied the dynamics of αS in the different coacervates by site-directed spin labeling (SDSL) continuous wave electron paramagnetic resonance (CW-EPR). This technique has proven useful to report on the flexibility and dynamic properties of IDPs with practically residue resolution[36–38]. To this end, we engineered cysteine residues in single-Cys mutants and labeled them with a maleimide derivative of the spin probe 4-hydroxy-2,2,6,6-tetramethylpiperidine-N-oxyl (TEMPOL). More specifically, we introduced a TEMPOL probe at position 122 or at position 24 of αS (TEMPOL-122-αS and TEMPOL-24-αS). In the first case, we targeted the C-terminal region of the protein, which is involved in the interaction with the poly-cations. Conversely, position 24 could give us information on the overall dynamics of the protein inside the condensate. In both cases, the obtained EPR signal of the protein in the dispersed phase is consistent with a nitroxide radical in the fast motion regime. Upon phase-separation in presence of either Tau or pLK (100 μM TEMPOL-αS at a

1:1 ratio for Tau441 or ΔN$_t$-Tau, or 1:10 for pLK), the EPR spectra of αS show a loss of peak intensities associated to line broadening, which indicates a reduction of the αS reorientation dynamics in the liquid droplets as compared to the protein in the diluted phase (Fig. 3d, Supplementary Fig. 4a). These changes are more pronounced at position 122. While at position 24 the dynamics of the probe is not affected by the presence of pLK, at position 122 there is a significant change in the line shape of the spectrum (Supplementary Fig. 4a). When we tried to simulate the spectra of position 122 for the two αS/poly-cation systems using the isotropic model (Supplementary Fig. 5a), commonly used to describe the dynamics of spin-labeled IDPs[38,39], we could not recover the experimental spectra, in contrast to the simulations of the spectra with the spin at position 24 (Supplementary Fig. 5a). This suggests that there are preferred locations in the configurational space of the spin at the αS C-terminal region in the presence of poly-cations. When taking into account the fraction of αS in the condensed phase under the EPR experimental conditions (84 ± 2%, 79 ± 7% and 47 ± 4% for αS/Tau441, αS/ΔN$_t$-Tau and αS/pLK, respectively—see data analysis in Supplementary Figure 2e), it becomes evident that the broadening detected by EPR reflects primarily the interaction of the αS C-terminal region with the various poly-cations in the condensed phase (major changes when TEMPOL-122-αS is used), rather than an increase in micro-viscosity experienced by the probe upon protein condensation. As expected, when adding 1 M NaCl to the mixture, the EPR spectrum of the protein in non-LLPS conditions is completely recovered (Supplementary Fig. 4b). Overall, our data indicate that the changes detected by CW-EPR reflect primarily the interaction of the C-terminal region of αS with the various poly-cations in the condensed phase and that this interaction seems to be stronger with pLK than Tau.

To obtain more structural information of the protein inside the coacervates, we set out to study the LLPS system by solution-state NMR. However, we could only detect the fraction of αS that remains in the dispersed phase, likely due to a combination of the reduced protein dynamics inside the coacervates and the deposition of the dense phase at the bottom of the solution within the experimental time required for the NMR analysis. When we analyzed the structure and dynamics of the protein that remains in the dispersed phase in the LLPS samples by NMR (Supplementary Fig. 5c, d), we observed that the protein behaves almost identically in the presence of pLK and ΔN$_t$-Tau, both in terms of secondary structure and protein backbone dynamics, as detected by secondary chemical shifts and $R_{1ρ}$ relaxation experiments. The NMR data show that the C-terminus of αS undergoes the main loss of conformational flexibility, while maintaining the disordered nature, as the rest of the protein sequence, due to its interaction with the poly-cations.

Since the broadening of the CW-EPR signal observed in TEMPOL-122-αS in the condensed phase reflects the protein's interaction with the poly-cations, we performed EPR titrations to assess the binding affinity of αS for the different poly-cations in the absence of LLPS (LLPS buffer without crowder), assuming that the interaction would be the same in the diluted and condensed phase (as supported by our data, Supplementary Fig. 4a and Supplementary Fig. 6). The aim was to observe whether, despite the common liquid-like nature of all coacervates, they show any underlying differential behavior at a molecular scale. As expected, upon increasing the poly-cation concentration, the EPR spectrum broadens, reflecting a decrease in molecular flexibility due to molecular interactions up to a near-saturation regime for all interacting partners (Fig. 3e, Supplementary Fig. 6). This saturation is achieved at a lower molar ratio (poly-cation:αS) for pLK, as compared to ΔN$_t$-Tau and Tau441. Indeed, fitting the data with an approximated binding model assuming $n$-identical and independent binding sites revealed an apparent dissociation constant an order of magnitude lower for pLK (~5 μM) than for Tau441 or ΔNt-Tau (~50 μM). Despite being a rough estimate, this indicates a higher affinity of αS for simpler

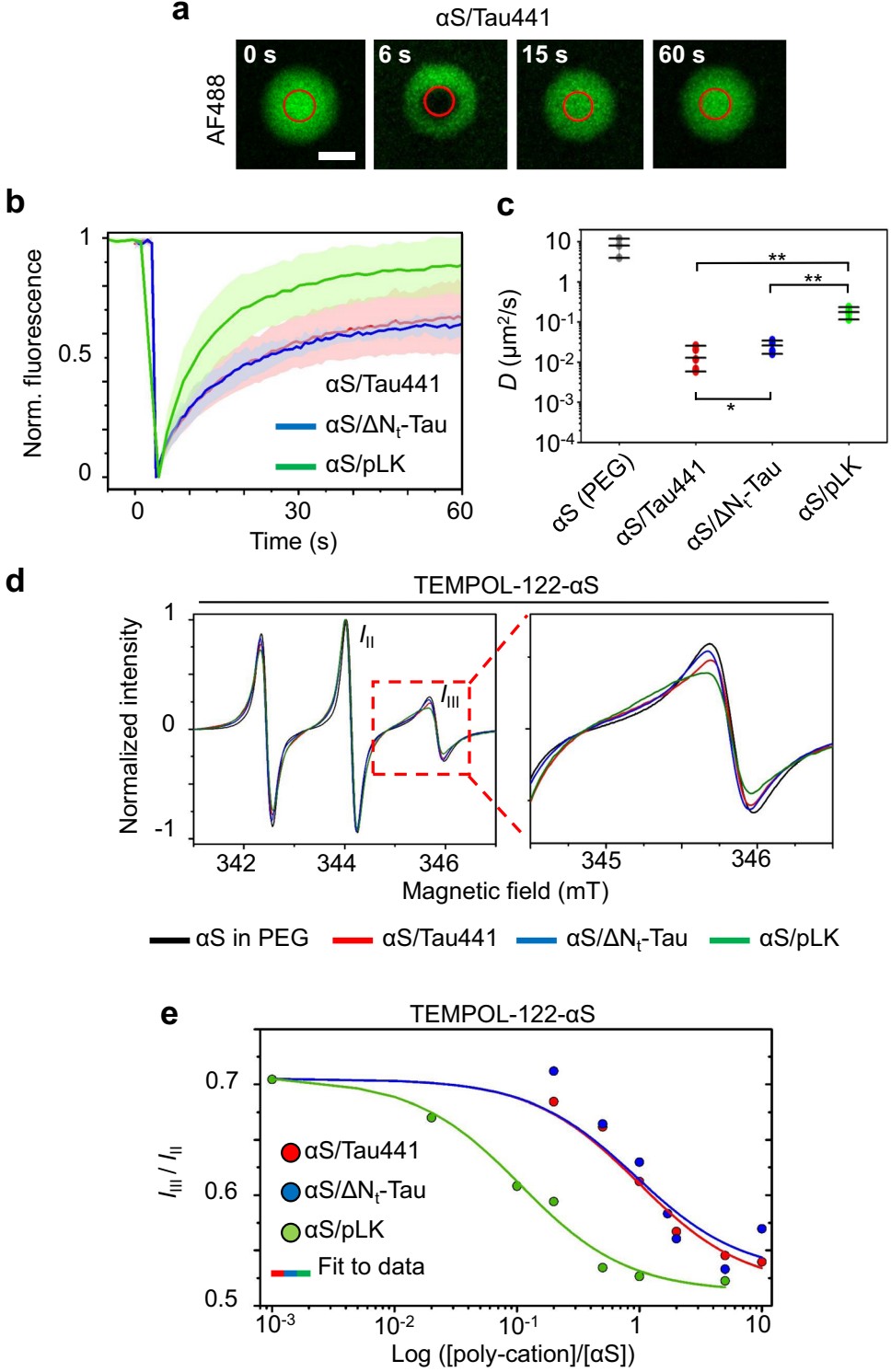

**Fig. 3 | αS dynamics in electrostatic complex coacervates. a–c** FRAP analysis of αS dynamics (2% AF488-labeled αS) within electrostatic coacervates. Representative images of a triplicate αS/Tau441 FRAP assay are shown in (**a**), where the red circle indicates the bleached area. Scale bar is 5 µm. **b** Mean FRAP curves and (**c**) calculated diffusion coefficients (*D*) of 5–6 (N) different droplets from a triplicate experiment with 100 µM αS and equimolar concentrations of Tau441 (red) or ΔNt-Tau (blue) or tenfold concentration of pLK (green) under LLPS conditions. The standard deviations of the FRAP curves are shown as shaded colors. For comparison, the diffusion coefficient of αS in the dispersed phase was determined in triplicate by fluorescence correlation spectroscopy (FCS) (see Supplementary Fig. 3 and methods for more information). **d** CW X-Band EPR spectra of 100 µM TEMPOL-122-αS in LLPS buffer without any poly-cation (black) or in the presence of a 100 µM Tau441 (red) or ΔNt-Tau (blue), or 1 mM pLK (green). The inset shows a zoom into the high-field line, where the most significant changes occur. **e** Binding curves of 50 µM TEMPOL-122-αS to the different poly-cations in the absence of LLPS (no PEG). The decrease in amplitude of band III relative to Band II (*I*III/*I*II) of normalized EPR spectra are shown for increasing molar ratios of Tau441 (red), ΔNt-Tau (blue) and pLK (green). Colored lines show the fit to data using an approximate binding model with n-identical and independent binding sites for each curve. Source data are provided as a Source Data file.

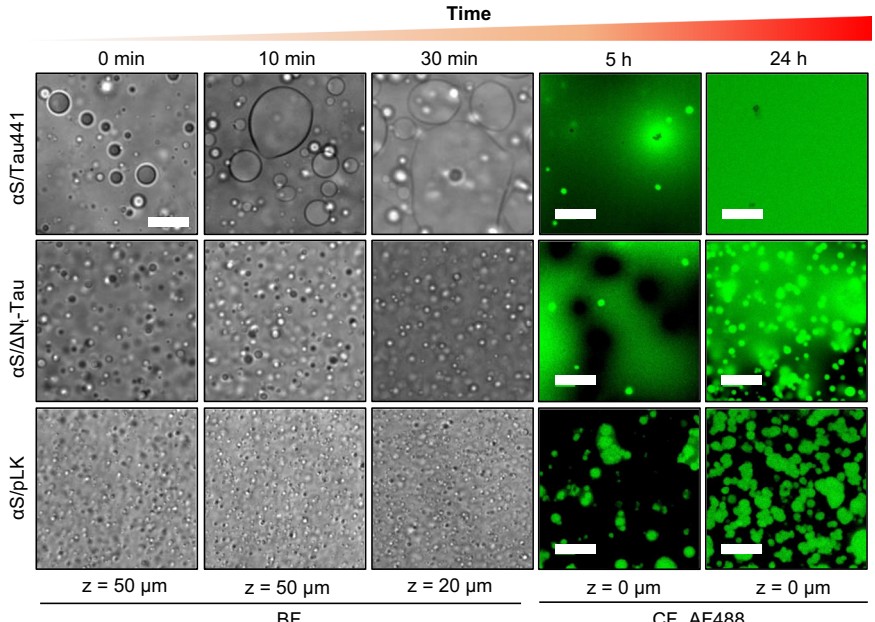

**Fig. 4 | Coalescence and wetting properties of αS electrostatic complex coacervates.** Representative BF (grayscale panels) and CF (right, fluorescence of AF488-labeled αS in green) microscopy images of coacervate samples with 100 μM αS (1% fluorescently-labeled) in LLPS buffer in the presence of 100 μM Tau441 (top), ΔN$_t$-Tau (middle) or 1 mM pLK (bottom) at different incubation times and focal heights (z, distance from the bottom of the plate well). 4–6 experiments were repeated independently with similar results. αS/Tau441 coacervate wetting after 24 h forms rafts larger than the image. Scale bar is 20 μm for all images.

poly-cations with an uninterrupted stretch of positive charges. Given such differences in affinities between αS and the various poly-cations, we hypothesized that their liquid properties might evolve differently over time, thus suffering different LSPT processes.

### Rapid gelation or slow LSPT-driven amyloid aggregation in αS/pLK vs αS/Tau441 coacervates

Considering the highly crowded environment inside the protein coacervates and the amyloidogenic nature of the proteins, we monitored the behavior of the coacervates over time in order to detect possible LSPT processes. By using BF and CF microscopy (Fig. 4), we observed that αS/Tau441 coacervates undergo fusion to a great extent in solution forming large droplets, which get in contact with the surface at the bottom of the well/slide and wet the surface, as expected for totally liquid droplets (Supplementary Fig. 7d); we termed these structures generated at the bottom "protein rafts". These structures remain liquid as they retain their fusion ability (Supplementary Fig. 7b) and can be seen within a few hours after LLPS is triggered (Fig. 4 and Supplementary Fig. 7c). We observed that the wetting process is favored on the surface of hydrophilic materials, but not hydrophobic ones (Supplementary Fig. 7a), as is expected for electrostatic coacervates with unbalanced charges and thus high electrostatic surface potential. Remarkably, coalescence and raft formation are significantly reduced for αS/ΔN$_t$-Tau and dramatically lower for αS/pLK condensates (Fig. 4). At short incubation times, αS/pLK droplets are able to fuse and wet hydrophilic surfaces, but this process is rapidly arrested and only limited fusion events and wetting-incompetent droplets are observed at incubation times longer than 5 h, indicative of a rapid liquid-to-gel droplet transformation.

Next, we wondered whether the enormous liquid-like protein reservoirs generated in αS/Tau441 LLPS would lead to amyloid aggregation of any of the investigated proteins. We monitored the maturation of αS/Tau441 droplets over time by WF microscopy under the same conditions as those described above, but with 1 μM AF488-labeled αS and Atto647N-labeled Tau441 (Fig. 5a). As expected, we observed total co-localization of the proteins at all times of the maturation process. Interestingly, from ca. 5 h onward, more intense,

non-round structures which we termed "puncta" were observed inside the rafts, some co-localizing with αS and some enriched in Tau441 (Fig. 5a white arrowheads). These puncta were always observed inside the rafts and to a greater extent for αS/Tau441 than αS/ΔN$_t$-Tau. No puncta were evident in fusion/wetting-incompetent droplets of both pLK and Tau systems. To test whether these puncta containing αS and Tau441 were amyloid-like aggregates, we conducted analogous experiments by CF microscopy where Tau441 was labeled with Atto647N and adding 12.5 μM thioflavin-T (ThT), an amyloid-specific dye, to the solution from the beginning. While no ThT staining of αS/Tau441 droplets or rafts was observed even after 24 h incubation (Fig. 5b, top row—remaining droplets above protein rafts), ThT-positive structures, containing Atto647N-Tau441, inside the rafts were evident, which recapitulate the size, shape and location of the previously described puncta (Fig. 5b middle and bottom row), suggesting that the puncta might correspond to amyloid-like aggregates generated inside the aged liquid coacervates.

In order to investigate the changes in the coacervate protein networks during the liquid-to-solid transition in more detail, we used fluorescence lifetime imaging (FLIM) and Förster resonance energy transfer (FRET) microscopy (Fig. 6 and Supplementary Fig. 8 and 9). We hypothesized that coacervate maturation into more condensed or even solid-like aggregated protein structures would bring proteins and, in turn, their attached fluorescent probes, into closer contact, thus likely exerting a quenching effect manifested as a reduced lifetime (τ) of the probes as described previously[40–42]. Likewise, this decrease in τ could also be concomitant, for doubly labeled samples (AF488 and Atto647N as donor and acceptor FRET dyes, respectively), with an increase in FRET efficiency (E) upon condensation and LSPT of the coacervates. We monitored the raft and puncta formation of αS/Tau441 and αS/ΔN$_t$-Tau LLPS samples over time (at 25 μM each protein with 1 μM AF488-labeled αS and/or Atto647N-labeled Tau441 or ΔN$_t$-Tau in LLPS buffer). We observed a general trend consisting of a slight decrease in the fluorescence lifetimes of both AF488 ($\tau_{488}$) and Atto647N ($\tau_{647N}$) probes upon coacervate maturation (Fig. 6 and Supplementary Fig. 8c). Interestingly, this change was dramatically enhanced for the puncta inside the rafts (Fig. 6c), suggesting that

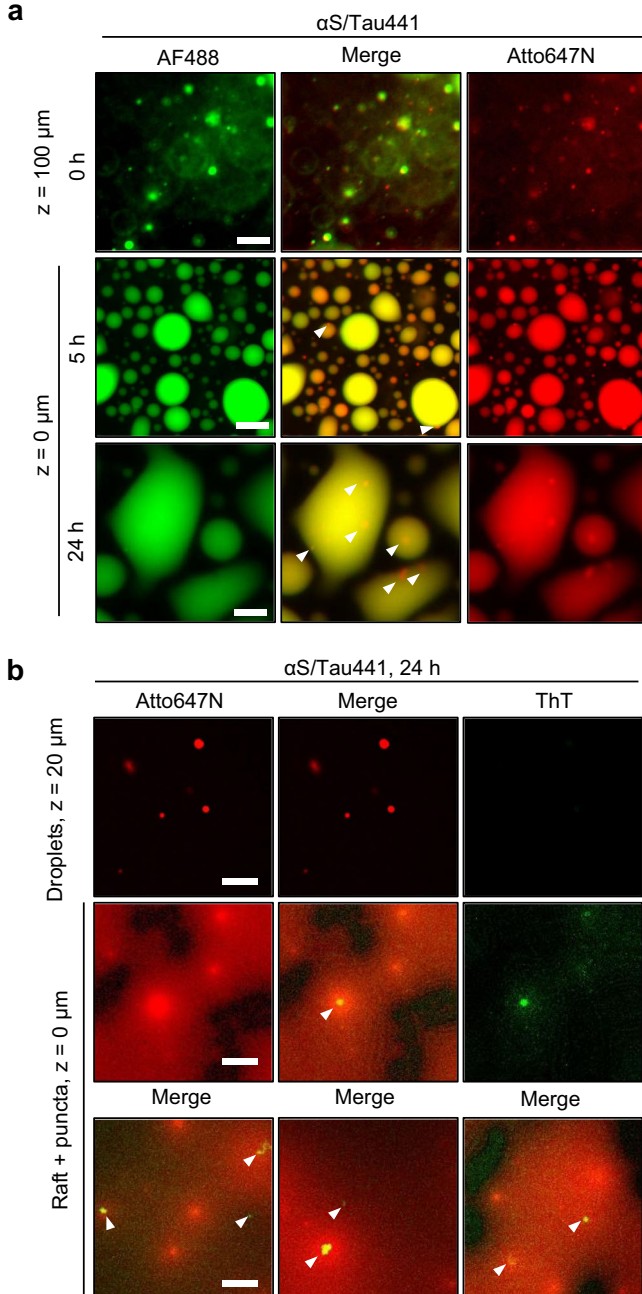

**Fig. 5 | Liquid-to-solid phase transition and amyloid aggregation of αS/Tau441 coacervates. a** WF microscopy images of 25 μM αS in the presence of 25 μM Tau441 (1 μM AF488-labeled αS and Atto647N-labeled Tau441) in LLPS buffer at different incubation times and focal heights (z, distance from the bottom of the non-binding plate well). Six experiments were repeated independently with similar results. **b** CF microscopy images of 25 μM αS in the presence of 25 μM Tau441 (1 μM Atto647N-labeled Tau441) and 12.5 μM thioflavin-T (ThT). Suspended protein droplets and deposited protein rafts and puncta are shown in the top and middle rows, respectively. The bottom row shows images of rafts and puncta from 3 independent replicate experiments. White arrowheads indicate ThT-positive puncta in both panels. Scale bar is 20 μm for all images.

further protein condensation occurs in the puncta. In support of this, no significant change in fluorescence lifetime was observed for 24 h-aged droplets of αS/ΔNₜ-Tau (Supplementary Fig. 8d), suggesting that droplet gelation is a process distinct from puncta formation and is not accompanied by a significant molecular reconfiguration within the coacervates. It needs to be mentioned that puncta had different sizes, with variable content in αS, particularly for the αS/Tau441 system

(Supplementary Fig. 8e). The reduced fluorescence lifetime in the puncta was concomitant with increased intensity, particularly for Atto647N-labeled Tau441 (Supplementary Fig. 8a), and higher FRET efficiency values, for both αS/Tau441 and αS/ΔNₜ-Tau systems, suggesting a further condensation of the proteins in the interior of the electrostsatic coacervates already at 5 h after LLPS was triggered. We observed lower $\tau_{647N}$ and slightly higher $\tau_{488}$ values, concomitant with lower and more heterogeneous FRET values, in αS/Tau441 puncta compared to αS/ΔNₜ-Tau. Arguably, this might stem from the fact that in the αS/Tau441 system a more heterogeneous content of αS, generally sub-stoichiometric with respect to Tau, is observed and expected in the aggregates, as Tau441 can also undergo LLPS and aggregation by itself (Supplementary Fig. 8e). The extent of droplet coalescence, raft formation and, importantly, protein aggregation inside liquid-like coacervates is, however, maximized when both Tau441 and αS are present.

To further prove the amyloid-like nature of the puncta/aggregates, we treated stain-free 24 h-aged coacervate samples with high concentrations (1 M) of NaCl, which resulted in the isolation of the aggregates from the protein coacervates. When the isolated aggregates (i.e., dispersed solutions of aggregates) were visualized by atomic force microscopy (AFM), we observed mostly globular-like morphologies, with regular heights of around 15 nm, that tend to associate under conditions of high salt concentrations, similar to the behavior of typical amyloid fibrils, due to a high hydrophobic surface exposure (note that the fibrils have typically heights of ca. 10 nm) (Supplementary Fig. 10a). Interestingly, when the isolated aggregates were incubated with ThT in standard ThT fluorescence assays, we observed a remarkable increase in the ThT fluorescence quantum yield, comparable to that observed when the dye is incubated with canonical αS amyloid fibrils (Supplementary Fig. 10b), suggesting that the coacervate-derived aggregates contain an amyloid-like structure. Indeed, the aggregates were resistant to high salt concentrations, but sensitive to 4 M guanidinium chloride (GdnHCl), like typical amyloid fibrils[43] (Supplementary Fig. 10c).

We further analyzed the composition of the aggregates by single-molecule fluorescence techniques, concretely fluorescence correlation/cross-correlation spectroscopy (FCS/FCCS) and two-color coincidence detection (TCCD) burst analysis. For this, we isolated the aggregates generated after 24 h incubation in 100 μL LLPS samples containing αS and Tau441 (both at 25 μM) with 1 μM AF488-labeled αS and 1 μM Atto647N-labeled Tau441. The resulted dispersed aggregate solutions were diluted to single-molecule conditions (typically a 1/500 dilution was required) with the same buffer but without PEG and with 1 M NaCl (the same buffer used to isolate the aggregates from the coacervates) in order to prevent both LLPS and any possible electrostatic interactions between the proteins. An example of single-molecule time trace can be observed in Fig. 7a. FCCS/FCS (cross-correlation, CC, and auto-correlation, AC) analyses indicated that aggregate containing both αS and Tau are highly represented in the sample (see CC curve in Fig. 7b, left panel), together with an excess of residual monomeric protein as a consequence of the processes of aggregate isolation and sample dilution (see AC curves in Fig. 7b, left panel). Control experiments with samples containing only monomeric proteins under the same solution conditions did not show CC curve and the AC curves were well fit to a one-diffusion component model (Eq. 4) with the diffusion coefficient expected for the monomeric proteins (Fig. 7b, right panel). The diffusion coefficients obtained for the aggregated particles were below 1 μm²/s, while those of the monomeric proteins were ca. 50–100 μm²/s; values similar to those previously reported for sonicated αS amyloid fibrils and monomeric αS under similar solution conditions, respectively[44]. When we analyzed the aggregates by TCCD burst analysis (Fig. 7c, upper panel), we found that ~60% of the detected aggregates contained both αS and Tau in each isolated

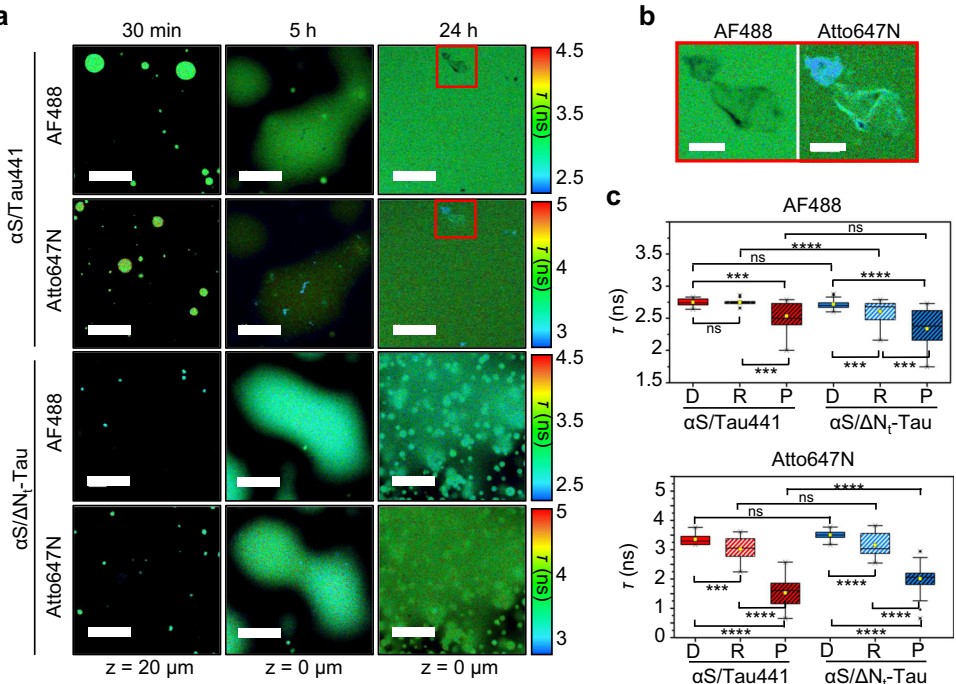

**Fig. 6 | Analysis of the liquid-to-solid phase transition of αS/Tau coacervates.**
**a** Fluorescence lifetime imaging microscopy (FLIM) images of αS/Tau441 and αS/ΔNt-Tau with 25 μM of each protein (1 μM AF488-labeled αS and 1 μM Atto647N-labeled Tau441 or ΔNt-Tau) in LLPS buffer. Columns show representative images of the LLPS samples at different maturation times (30 min, 5 h and 24 h). Red boxes show a region containing puncta for αS/Tau441. Lifetime range shown as a color scale. Scale bar = 20 μm for all images. **b** Zoom-in FLIM image of the selected regions shown in red boxes in panel **a**. Lifetime range shown with the same color scale as in panel **a**. Scale bar =5 μm. **c** Box plots showing the lifetime distributions of AF488 (attached to αS) or Atto647N (attached to Tau) for the different protein species (droplets -D-, rafts -R- and puncta -P-) identified in the FLIM images

recorded for αS/Tau441 and αS/ΔNt-Tau coacervate samples ($N$ = 17–32 ROIs for D, 29–44 ROIs for R and 21–51 ROIs for puncta). Mean and median values are shown as yellow squares and black lines within the boxes, respectively. Lower and upper box limits indicate the first and third quartile, respectively, while minimum and maximum values within 1.5 × interquartile range (IQR) are shown as whiskers. Outliers are shown as black diamonds. The statistical significance between pairs of distributions was determined with a two sample $t$-test assuming unequal variances. The $p$ values from two-tailed $t$-test are shown as stars for each compared pair of data (*$p$ value > 0.01, **$p$ value > 0.001, ***$p$ value > 0.0001, ****$p$ value > 0.00001), ns means not significant ($p$ value > 0.05). Precise $p$ values are provided in Supplementary Table 1 and source data as a Source Data file.

aggregate (αS/Tau hetero-aggregates), ~30% contained only Tau, and ~10% only αS. The analysis of the stoichiometries of the αS/Tau hetero-aggregates showed that the majority of the hetero-aggregates were enriched in Tau (stoichiometries below 0.5, with an average of ca. 4 times more Tau than αS molecules per aggregate), in agreement with what we observed in in-situ experiments by FLIM. The FRET analysis demonstrated that these aggregates contain both proteins, although the actual FRET value does not have major interest in this case, as the fluorophores distribution in each aggregate is random due to the excess of unlabeled protein used in the experiments. Interestingly, when we performed the same analysis using a well-established amyloid aggregation deficient Tau variant[45,46] (see Supplementary Fig. 11a, b), we observed that while the electrostatic coacervation with αS was identical to the full-length Tau variant (Supplementary Fig. 11c, d), the ability to form aggregates inside the coacervates was drastically reduced and almost no puncta was detected by FLIM in in-situ experiments, and a faint cross-correlation curve was observed for the isolated aggregates sample. For the small number of aggregates detected (only one-tenth as many as with Tau441), however, we observed an enrichment in αS over this Tau variant per aggregate, with ~50% of the aggregates detected containing only αS molecules, and an excess of αS in the hetero-aggregates (see Supplementary Fig. 11e), in contrast to the hetero-aggregates generated with Tau441 (Fig. 6f). The results of these experiments demonstrate that while αS is capable of aggregating per se inside the coacervates with Tau, Tau nucleation is more favorable under these conditions, generating amyloid-like aggregates able to act as nuclei for the formation of hetero-aggregates of

αS and Tau. Once the Tau-rich nuclei however is formed, the heterotypic interactions between αS and Tau are favored in the aggregates over the homotypic interactions between Tau molecules; a characteristic that we also observed in the protein networks of αS/Tau liquid coacervates.

## Discussion

Maturation or ageing of liquid protein condensates over time into gel-like or solid-like structures has been reported to be relevant for the functioning of certain physiological condensates[47], but also to disease, as an aberrant process that precedes amyloid aggregation[7,48,49]. Here, we present a detailed study of the phase-separating and LSPT behavior of αS in the presence of disordered poly-cations in a controlled environment at low micromolar concentrations and physiologically relevant conditions (note that the estimated physiological concentration of αS is >1 μM[50]), following the typical thermodynamically-driven LLPS behavior. We have found that αS, containing a highly negatively charged C-terminal region at physiological pH, is able to form protein-rich liquid droplets in aqueous solutions by LLPS in the presence of highly cationic disordered polypeptides, such as pLK or Tau, by a process of electrostatic complex coacervation, in the presence of macromolecular crowding. This process could have relevant implications in a cellular context, where αS encounters a variety of poly-cationic molecules that have been related to its disease-linked aggregation both in vitro and in vivo[51–54].

In a number of studies, protein dynamics within liquid droplets have been suggested as one of the key factors dictating the maturation process[55,56]. In the electrostatic coacervates of αS with

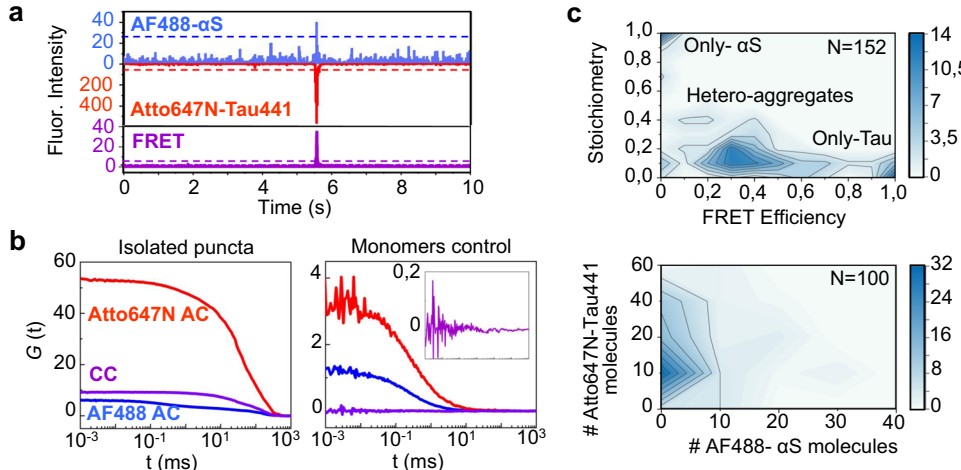

**Fig. 7 | Analysis of the isolated LLPS-derived αS/Tau aggregates by single-molecule fluorescence. a** A representative single-molecule fluorescence time trace of isolated aggregates generated inside αS/Tau441 electrostatic coacervates. A burst corresponding to an αS/Tau441 co-aggregate (burst above the assigned thresholds) is observed in the three detection channels (emission of AF488 and Atto647N after direct excitation, blue line and red lines, respectively, and emission of Atto647N after indirect excitation, FRET, violet line). **b** FCS/FCCS analysis of an isolated LLPS-derived αS/Tau441 aggregate sample (left panel). The auto-correlation (AC) curves for AF488 and Atto647N are shown in blue and red, respectively, and the cross-correlation (CC) curve, related to the aggregates containing both dyes, is shown in violet. The AC curves reflect the presence of both labeled monomeric and aggregated protein species, while the CC curve only shows the diffusion of dual-labeled aggregates. The same analysis but with a sample containing only monomeric αS and Tau441 under the same solution conditions as the isolated puncta is shown in the right panel as control. **c** Single-molecule fluorescence burst analysis of isolated aggregates generated inside αS/Tau441 electrostatic coacervates. The information of each aggregate detected in four different replicas ($N = 152$) was plotted according to their stoichiometry, S, and FRET efficiency value (upper panel, the color scale reflects occurrence). Three types of aggregates can be distinguished: only-αS aggregates, with S~1 and FRET~0, only-Tau aggregates, with S ~ 0 and FRET ~ 1, and Tau/αS hetero-aggregates, with intermediate S and FRET values. An estimation of the number of the two labeled-proteins in each hetero-aggregate detected ($N = 100$) is shown in the bottom panel (the color scale reflects occurrence). Source data are provided as a Source Data file.

poly-cations, the maturation process seems to be governed by the strength of the interaction with the poly-cation and the valence and multiplicity of these interactions. Equilibrium theories establish that the equilibrium landscape of two liquid phase states would be the presence of one large droplet rich in the biopolymers that drive the LLPS[57,58]. The growth of liquid droplets could be achieved by Ostwald ripening[59], coalescence[60] or by consumption of free monomers from the disperse phase[61]. In the case of αS and Tau441, ΔN$_t$-Tau or pLK, most of the protein is concentrated in the condensates under the conditions used in this study. However, while full-length Tau droplets rapidly undergo coalescence accompanied by surface wetting, droplet fusion and wetting is hindered for ΔN$_t$-Tau and, to a dramatic extent, for pLK, indicative of a rapid loss of liquid properties in these two systems. According to our FLIM-FRET analysis, the aged pLK and ΔN$_t$-Tau droplets show a similar degree of protein condensation as the initial droplets (similar fluorescence lifetimes), indicating that the initial protein network is preserved, although it becomes more rigid.

We have rationalized our experimental results in the following model (Fig. 8). Initially, the instantaneously formed droplets generally present a protein network that is not electrostatically compensated so that there are charge unbalanced regions, particularly at the droplet interface, resulting in droplets with high electrostatic surface potential. In order to compensate the charges (phenomenon generally referred to as valence exhaustion) and minimize the droplet surface potentials, droplets can incorporate new polypeptides from the diluted phase, re-organize the protein network to optimize charge-charge interactions, fuse with other droplets or interact with the surface (wetting). αS/pLK droplets, due to the simpler protein network (only heterotypic interactions between αS and pLK) and the stronger affinity of the protein-protein interactions, seem to be able to balance the charges of the condensates faster; indeed, we have observed faster protein dynamics in the initially formed αS/pLK coacervates than in those of αS/Tau. Upon valence exhaustion, the interactions become less transient and the droplets lose liquid properties, converting into gel-like, fusion-incompetent droplets with low electrostatic surface potential (thus unable to wet surfaces). In contrast, the αS/Tau droplets, are less efficient in optimizing the charge balance of the droplets as a consequence of the more complex protein network (with both homotypic and heterotypic interactions) and the weaker nature of the protein interactions. This results in droplets that keep a liquid-like behavior for longer times, and present high electrostatic surface potentials, which tend to be minimized by coalescing and growing (thus minimizing the droplets surface area/volume ratio), as well as by wetting hydrophilic surfaces. This generates large condensed protein reservoirs that maintain the liquid nature as there is a continuous search for the optimization of the charges in the protein network and thus the interactions remain highly transient. Interestingly, N-terminally truncated forms of Tau, including some natural isoforms[62], would show an intermediate behavior, with some coacervates with αS aging into long-lived, gel-like droplets, while others evolving into large liquid-like condensates. This duality in the maturation of the αS electrostatic coacervates is in line with recent theoretical and experimental LLPS studies that have highlighted the relevance of valence exhaustion and electrostatic screening within condensates as a key mechanism to control the size of the condensates and their liquid properties[58,61].

The resulting large, liquid-like structures, with a highly crowded but dynamic protein environment, formed during the maturation of αS/Tau441 and, to a lesser extent, αS/ΔN$_t$-Tau coacervates, are ideal reservoirs for the nucleation of protein aggregation. We have indeed observed the formation of solid protein aggregates in such type of protein coacervates, which typically contain both αS and Tau. We have demonstrated that these hetero-aggregates are stabilized by non-electrostatic interactions, are able to bind the amyloid-specific ThT dye in the same way as canonical amyloid fibrils do and, indeed, share similar stabilities with respect to different treatments, suggesting that the αS/Tau aggregates formed by LLPS have an amyloid-like nature. Indeed, a well-established amyloid aggregation deficient Tau variant is

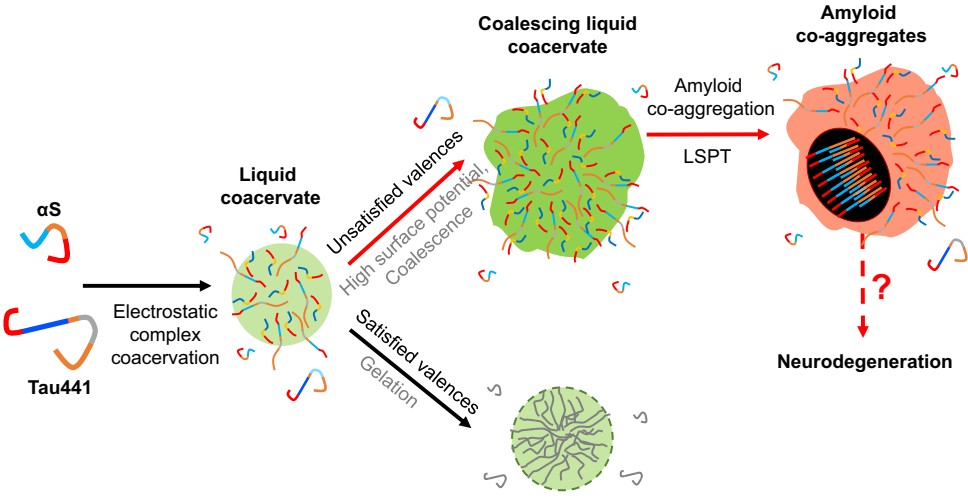

**Fig. 8 | Model for the synchronized LLPS and LSPT-driven amyloid hetero-aggregation of αS and Tau441.** The schematic shows the proposed amyloid aggregation pathway for αS and Tau441 through LLPS and LSPT. With complementary anion-rich (red) and cation-rich (blue) regions, αS and Tau electrostatic coacervates with satisfied valences have a lower surface energy and thus coalesce to a lesser extent, allowing the droplet to age rapidly reaching a stable, non-coalescing gel-like state. This situation is highly favorable in the case of the αS/pLK system owed to a higher affinity and simpler one-pair protein interaction network, which allows for a rapid gel-like transition. In contrast, droplets with unsatisfied valences and, thus, interaction-available protein charged regions, will make the coacervates more prone to coalesce and wet hydrophilic surfaces in order to reduce their high surface energy. This situation is preferred in the αS/Tau441 coacervates, which have a multivalent complex network composed of Tau-Tau and αS-Tau weak interactions. Larger coacervates, in turn, will be more susceptible to retain their liquid-like properties, allowing for other protein-protein interactions to occur. Eventually, amyloid hetero-aggregates containing both αS and Tau are formed within the liquid coacervates, which might be related to those found in neurodegenerative disease-hallmark inclusions.

remarkably impaired to form these hetero-aggregates with αS in the interior of the liquid electrostatic coacervates. The αS/Tau441 aggregates are only observed to form inside the coacervates that maintain liquid-like properties, and have never been observed if the coacervates/droplets have reached a gel-like state. In the latter case, the increase in strength of the electrostatic interactions and thus the concomitant rigidity of the protein network impedes the necessary protein conformational rearrangements for establishing the new type of protein interactions required for amyloid nucleation. This could be, however, achieved in a more flexible, liquid-like coacervate, which, in turn, is more likely to remain liquid upon increasing its size.

The fact that aggregate formation inside the condensed phase is chiefly favored in large αS/Tau condensates over small droplets, which can rapidly suffer gelation, highlights the relevance of identifying the factors that control droplet fusion. It is, therefore, not just the tendency to phase separate but also the size of the condensates that must be regulated for their proper functioning as well as for disease prevention[58,61]. Our results also emphasize the importance of the balance between LLPS and LSPT for the αS/Tau system. While droplet formation might play a protective role regarding amyloid aggregation by reducing the amount of available protein monomers in saturating conditions, as it has been already proposed in other systems[63,64], droplet coalescence of highly liquid droplets might lead to protein aggregation in the interior of the coacervates by a slow conformational rearrangement of the protein network.

Overall, our data strongly underline the relevance of coacervate valence and satisfied/unsatisfied interactions in the droplet network in the context of LSPT. In particular, we show that full-length αS/Tau441 condensates are capable of effectively coalescing and nucleating to form amyloid-like hetero-aggregates involving both proteins, and propose a molecular mechanism based on our experimental results. The co-aggregation of both proteins inside the αS/Tau liquid coacervates that we report here could indeed be related to the co-localization of both proteins in disease-hallmark inclusions and might pave the way for understanding the link between LLPS and amyloid aggregation of highly charged IDPs in neurodegeneration.

## Methods

### Protein expression, purification and labeling

Monomeric WT-αS, the cysteine mutants (Q24C-αS, N122C-αS) and the ΔCₜ-αS variant (Δ101–140) were expressed in *Escherichia coli* and purified as described before[21]. 5 mM DTT was included in all purification steps for the cysteine αS mutant variants to prevent disulfide bridge formation. Tau441 isoform (plasmid obtained from Addgene #16316), the ΔNₜ-Tau variant (Δ1–150, generated by IVA cloning with primers CTTTAAGAAGGAGATATACATATGATCGCCACACCGCGG, CATATGTATATCTCCTTCTTAAAGTTAAAC) and the AggDef-Tau variant (Δ275–311, primers CTTCCCGCCTCCCGGCTGG, CCGGGAGGCGGGAAGCCAGTT-GACCTGAGCAAGGTGACCT) were purified as described in[65] with the following modifications: *E. coli* cultures were grown at 37 °C and 180 rpm to an $OD_{600} = 0.6$–0.7 and expression was induced with IPTG at 37 °C for 3 h. Cells were harvested at 11,500 × *g* for 15 min at 4 °C and washed with saline buffer containing NaCl 150 mM. The pellet was resuspended in lysis buffer (20 mL per 1 L of LB: MES 20 mM pH 6.8, NaCl 500 mM, EDTA 1 mM, MgCl₂ 0.2 mM, DTT 5 mM, PMSF 1 mM, benzamidine 50 μM, leupeptin 100 μM). The sonication step was performed on ice at 80% amplitude with 10 pulses (1 min on, 1 min off). No more than 60 mL were sonicated at once. *E. coli* lysate was heated at 95 °C for 20 min and then cooled on ice and centrifuged for 40 min at 127,000 × *g*. The cleared supernatant was loaded into a 3.5 kDa membrane (Spectrum™ Thermo Fisher Scientific, UK) and dialyzed against 4 L of dialysis buffer (MES 20 mM pH 6.8, NaCl 50 mM, EDTA 1 mM, MgCl₂ 2 mM, DTT 2 mM, PMSF 0.1 mM) for 10 h. A 5 mL cationic exchange column (HiTrap SPFF, Cytiva, MA, USA) was equilibrated with equilibration buffer (MES 20 mM pH 6.8, NaCl 50 mM, EDTA 1 mM, MgCl₂ 2 mM, DTT 2 mM, PMSF 0.1 mM). Tau lysate was filtered through a 0.22 μm PVDF filter and injected into the column at a flow rate of 1 mL/min. Elution was performed gradually, and Tau was eluted at 15–30% elution buffer (MES 20 mM pH 6.8, NaCl 1 M, EDTA 1 mM, MgCl₂ 2 mM, DTT 2 mM, PMSF 0.1 mM). Fractions were analyzed by SDS-PAGE, and all the fractions containing a single band at the expected MW of Tau were concentrated together with a 10 kDa centrifugal filter and changed to a buffer containing HEPES 10 mM, pH 7.4, NaCl 500 mM and DTT

2 mM to a final protein concentration of 100 μM. The protein solution was then passed through a 0.22 μm PVDF filter, flash-frozen and stored at −80 °C. K18 protein was kindly provided by Prof. Alberto Boffi. Purity of the preparations was >95% as confirmed by SDS-PAGE and MALDI-TOF/TOF. Labeling of the different cysteine αS variants by maleimide chemistry with either AlexaFluor488-maleimide (AF488, ThermoFisher Scientific, Waltham, MA, USA) or TEMPOL-maleimide (Toronto Research Chemicals, Toronto, Canada) was performed as described before[21] and the degree of labeling (>95%) was confirmed by absorbance and MALDI-TOF/TOF. Tau441, ΔN$_t$-Tau, AggDef-Tau and K18 were labeled with Atto647N-maleimide (ATTO-TEC GmbH, Siegen, Germany) using the natural cysteine residues at positions 191 and 322, following the same procedures. The net charge per residue diagram of αS and Tau441 was obtained using CIDER[66].

## Liquid-liquid phase separation (LLPS) assays

Solid poly-L-lysine (pLK with a degree of polymerization 90–110 as determined by NMR from the vendor, Alamanda Polymers Inc, Huntsville, AL, USA) was dissolved in 10 mM HEPES, 100 mM NaCl, pH 7.4 to a concentration of 10 mM, sonicated for 5 min in a bath sonicator and stored at −20 °C. PEG-8, dextran-70, FITC-PEG-10 (Biochempeg, Watertown, MA, USA) and FITC-dextran-500 (Sigma-Aldrich, Sant Louis, MI, USA) were dissolved in water and dialyzed in LLPS buffer extensively to remove contaminant salts. They were further filtered through a 0.22 μm syringe filter and their concentration was calculated using a refractometer (Mettler Toledo, Columbus, OH, USA). LLPS samples were prepared at room temperature in the following order: Buffer and crowder were mixed and supplemented with 1 mM tris(2-carboxyethyl) phosphine (TCEP, Carbosynth, Compton, UK), 1 mM 2,2,2-(Ethane-1,2-diyldinitrilo) tetraacetic acid (EDTA, Carbosynth) and 1% protease inhibitor cocktail (PMSF 100 mM, benzamidine 1 mM, leupeptin 5 μM). Then, αS and the coacervating poly-cation (pLK or Tau variants) were added. For thioflavin-T (ThT, Carbosynth, Compton, UK) time-series experiments, a total ThT concentration was used so that it was half of the concentration of αS. Samples were mixed gently but thoroughly to ensure their homogeneity. Concentrations of each component varied among experiments as described in the results section. Azide was used at a 0.02% (w/v) concentration whenever an experiment lasted longer than 4 h. For all assays with LLPS samples, mixtures were allowed to equilibrate for 5 min before assaying. For light scattering assays, 150 μL samples were spotted onto Non-Binding 96-Well Microplates (μClear®, Black, F-Bottom/Chimney Well, Greiner bio-one, Kremsmünster, Austria) and the plate was covered with an adhesive foil. LLPS was monitored by measuring the absorbance at 350 nm at the center of the solution in a CLARIOstar plate reader (BMG Labtech, Ortenberg, Germany). Experiments were performed at 25 °C in triplicate and errors were calculated as the standard deviation from the mean. The quantification of the αS fraction in the diluted and condensed phases in the different LLPS solutions was carried out by sample centrifugation and quantification of the diluted phase by SDS-PAGE gel analysis. 100 μL LLPS samples containing 1 μM AF488-labeled αS were prepared by thorough mixing and subsequently centrifuged for 30 min at 9600 × g upon which a pellet was typically visible. The upper 50 μL of supernatant were used for protein quantification by SDS-PAGE gel. The gel was scanned with an AF488 filter using a ChemiDoc Gel Imaging System (Bio-Rad Laboratories, Hercules, CA, USA) or stained with coomasie stain and imaged with the corresponding filter. The resulting bands were analyzed using ImageJ 1.53i version (NIH, USA). Experiments were performed in duplicate in two different experiments with similar results.

## Brightfield (BF), Differential interference contrast (DIC) and widefield fluorescence (WF) microscopy

One hundred and fifty microliters samples were typically spotted onto Non-Binding 96-Well Microplates and imaged on a Leica DMI6000B inverted microscope (Leica Microsystems, Wetzlar, Germany) at room temperature. For punctual experiments, μ-Slide Angiogenesis dishes (Ibidi GmbH, Gräfelfing, Germany) or 96-well polystyrene microplates (Corning Costar Corp., Acton, Massachusetts) were also used. A halogen lamp or a mercury metal halide bulb EL6000 (for BF/DIC and WF imaging, respectively) served as illumination sources. For WF microscopy, the light was focused on and collected from the sample using a 40x air objective lens (Leica Microsystems, Germany). For AF488- and ThT-labeled samples, the excitation and emission light was filtered with a standard GFP filter set with bandpass filters of 460–500 nm and 512–542 nm for excitation and emission, respectively, and a dichroic mirror of 495 nm. For Atto647N-labeled samples, a standard Cy5 filter set was used with 628–40 nm and 692–40 nm excitation and emission bandpass filters, respectively, and a dichroic mirror of 660 nm. For BF and DIC microscopy, the same objective was used to collect the reflected light. Collected light was detected on a Leica DFC7000 CCD camera (Leica Microsystems, Germany). Exposure times were 50 ms for BF and DIC microscopy imaging and 20–100 ms for WF microscopy imaging. For comparative purposes, exposure time was 100 ms for all ThT experiments. For droplet fusion visualization, time-lapse experiments were performed, collecting images every 100 ms for several minutes. ImageJ (NIH, USA) was used for image analysis. Experiments were performed in triplicate with similar results.

## Confocal fluorescence (CF) microscopy

For co-localization, FRAP and 3D-reconstruction experiments, images were acquired on a Zeiss LSM 880 inverted confocal microscope using ZEN 2 blue edition (Carl Zeiss AG, Oberkochen, Germany). Fifty microliters samples were spotted onto a μ-Slide Angiogenesis dish (Ibidi GmbH, Gräfelfing, Germany) treated with a hydrophilic polymer (ibiTreat) and placed on top of a 63× immersion oil objective lens (Plan-Apochromat 63×/N.A. 1.4 Oil DIC). Images were acquired with a resolution of 0.26 μm/pixel and a dwell time of 8 μs/pixel using 458 nm, 488 nm and 633 nm argon laser lines for excitation and emission detection windows of 470–600 nm, 493–628 nm and 638–755 nm for ThT, AF488 and Atto647N imaging, respectively. For FRAP experiments, time-lapses of each sample were recorded at 1 frame-per-second. Experiments were performed at room temperature in triplicate with similar results. All images were analyzed using the software Zen 2 blue edition (Carl Zeiss AG, Oberkochen, Germany). FRAP curves were normalized, plotted and fitted using OriginPro 9.1 from intensity/time data extracted from the images with Zen 2. Recovery curves were fitted to a mono-exponential model to account for molecular diffusion with an additional exponential term to account for acquisition bleaching effects. Then, we calculated $D$ using the nominal bleaching radius and the recovery half-time previously determined, as in equation 5 of Kang et al.[35].

## Electron paramagnetic resonance (EPR)

Single cysteine αS variants were spin-labeled with 4-hydroxy-2,2,6,6-tetramethylpiperidine-N-oxyl (TEMPOL) at positions 24 (TEMPOL-24-αS) and 122, respectively (TEMPOL-122-αS). For EPR experiments, αS concentration was set to 100 μM and the concentration of PEG was 15% (w/v). For the different coacervation conditions, the αS:pLK ratio was of 1:10, while the ratios αS:ΔN$_t$-Tau and αS:Tau441 were kept to 1:1. For binding titration experiments in the absence of crowder, TEMPOL-122-αS was kept at 50 μM and the poly-cation was titrated in increasing concentrations, preparing each condition individually. CW-EPR measurements were performed with a Bruker ELEXSYS E580 X-band spectrometer equipped with a Bruker ER4118 SPT-N1 resonator operating at a microwave (MW) frequency of ~9.7 GHz. The temperature was set to 25 °C and controlled by a liquid nitrogen cryostat. Spectra were taken under non-saturating conditions with a MW power of 4 mW, a modulation amplitude of 0.1 mT and a modulation frequency of 100 kHz. Spectra intensity was normalized to avoid differences in

spin concentrations across samples and possible spin reductions due to residual concentrations of reducing agents in the samples containing Tau441 or $\Delta N_t$-Tau (present in the stock protein solutions). Reported g-values were obtained from simulations of EPR spectra performed with the Easyspin software (v. 6.0.0-dev.34) implemented in Matlab®[67]. A one/two-component isotropic model was used for simulating the data. After normalizing all the signals, residuals were calculated by subtracting each simulation from the corresponding experimental spectrum. For binding titration assays, the relative intensity of the third band to the second band of the normalized EPR spectra ($I_{III}/I_{II}$) was used to monitor the binding of the poly-cation to $\alpha$S. For estimating the dissociation constant ($K_d$), the resulting curves were fitted to an approximated model assuming n-identical and independent binding sites.

## Nuclear magnetic resonance (NMR)

NMR spectroscopy experiments were carried out utilizing a Bruker Neo 800 MHz ($^1$H) NMR spectrometer fitted with a cryoprobe and Z-gradients. All experiments used 130–207 μM $\alpha$S and the corresponding equivalents of $\alpha$S/$\Delta N_t$-Tau and pLK in 10 mM HEPES, 100 mM NaCl, 10% $D_2O$, pH 7.4, and were run at 15 °C. To monitor LLPS by NMR, 10% PEG was added to the premixed samples. Chemical shift perturbation plots (Fig. 1b) show the averaged $^1$H and $^{15}$N chemical shifts. The $\alpha$S 2D$^1$H-$^{15}$N HSQC spectrum was assigned based on previous assignments (BMRB entry #25227) and confirmed by recording and analysis of 3D HNCA, HNCO and CBCAcoNH spectra. $^{13}C_\alpha$ and $^{13}C_\beta$ chemical shifts were calculated in presence of $\Delta N_t$-Tau or pLK to measure possible changes in secondary structural trends upon comparison to the chemical shifts of $\alpha$S in a pure statistical coil conformation[68] (Supplementary Fig. 5c). $R_{1\rho}$ rates were measured by recording hsqctretf3gpsi experiments (obtained from the Bruker library) with delays of 8, 36, 76, 100, 156, 250, 400 & 800 ms and an exponential function was fit to the intensities of the peaks over the distinct time delays to determine the $R_{1\rho}$ rates and their experimental uncertainties.

## FLIM and FRET microscopy

Dual-color time-resolved fluorescence microscopy experiments were performed on a commercial MT200 (PicoQuant, Berlin, Germany) time-resolved fluorescence confocal microscope with a Time-Correlated Single Photon Counting (TCSPC) unit. Laser diode heads were used in Pulsed Interleaved Excitation (PIE) and the beams were coupled through a single-mode waveguide and adjusted to laser powers between 10 and 100 nW for both 481 nm and 637 nm laser lines measured after the dichroic mirror. This ensured optimal photon count rates while avoiding photon pile-up effects, photobleaching and saturation. The coverslip or μ-Slide Angiogenesis plate (Ibidi GmbH, Gräfelfing, Germany) was placed directly on the immersion water on top of a Super Apochromat 60x NA 1.2 objective with a correction collar (Olympus Life Sciences, Waltham, USA). A dichroic mirror of 488/640 nm (Semrock, Lake Forest, IL, USA) was used as the main beam splitter. Out-of-focus emission light was blocked by a 50 μm pinhole and the in-focus emission light was then split by a 50/50 beamsplitter into 2 detection paths. Bandpass emission filters (Semrock, Lake Forest, IL, USA) of 520/35 for the green dye (AF488) and 690/70 for the red dye (Atto647N) were used before the detectors. Single Photon Avalanche Diodes (SPADs) (Micro Photon Devices, Bolzano, Italy) served as detectors. Both data acquisition and analysis were performed on the commercially available software SymphoTime64 (PicoQuant GmbH, Berlin, Germany).

Fifty microliters LLPS samples were spotted onto a μ-Slide Angiogenesis well plate (Ibidi GmbH, Gräfelfing, Germany). Images were acquired focusing 20 μm above the well bottom for an optimal objective lens working distance for suspended droplets and at -1 μm for rafts and puncta, with an axial resolution of at least 0.25 μm/pixel and a dwell time of 400 μs/pixel. Data were selected by applying an intensity threshold based on the mean intensity of the background signal ($F_{BG,mean} + 2\sigma$) to each channel in order to select only liquid protein droplets, rafts or puncta, filtering out any dim signal which could originate from the dispersed phase. For species-specific lifetime ($\tau$) analysis of each channel (green, "g" for AF488 and red, "r" for Atto647N), we selected regions of interest (ROI) that contained either droplets, rafts or puncta (Supplementary Fig. 8b), and obtained their mean $\tau$ by fitting their lifetime decays ($\tau_D$, $\tau_R$ and $\tau_P$ for droplets, rafts or puncta, respectively, see Supplementary Fig. 8c) in each channel using the tail-fitting analysis and a 2-component decay model. ROIs that yielded too low photon counts for multi-exponential fitting were discarded from the analysis. The cutoffs used were < $10^4$ photons for rafts and puncta, and $10^3$ for droplets. The threshold for the droplets was lower since it was very difficult to obtain decay curves with higher intensity values because of the generally small size and reduced number of droplets per image field. ROIs with photon counts above the photon pile-up limit (set to > 500 counts/pixel) were also discarded for the analysis. Tail-fitting was performed on the ROI-derived intensity decay curves starting at a lifetime where intensity is 90% of the maximum (slightly after the decay's maximum intensity) to ensure minimal IRF interference while maintaining the same relative time window for all intensity decay fittings. Between 25 and 50 ROIs for rafts and puncta and 15–25 ROIs for droplets, selected from images obtained from more than 4 replicas recorded in at least 3 independent experiments were analyzed. Two-tailed t-tests were used for assessing statistical differences between the species or between coacervate systems. For pixel-wise lifetime ($\tau$) analysis, the overall lifetime decay of the entire field was calculated for each channel and was tail-fitted to a 2/3-component exponential decay model. Then, the lifetime decay of each individual pixel was fitted using the previously calculated $\tau$ values, yielding a FLIM-fitted false color-coded image. The lifetime range for tail-fitting was the same throughout all images of the same channel and enabled a sufficient amount of photons per decay to yield robust fits. For FRET analysis, pixels were selected by applying a lower intensity threshold of 100 photons, being the mean background signal ($F_{BG}$) 11 photons. The fluorescence intensity of each channel was corrected by the experimentally determined correction factors:[69] spectral cross-talk $\alpha$ was 0.004, direct excitation $\beta$ was 0.0305 and detection efficiency $\gamma$ was 0.517. Pixel-wise FRET efficiency was then calculated as given by the following equation:

$$E = \frac{F_{DA} - \alpha F_{DD} - \beta F_{AA}}{F_{DA} - \alpha F_{DD} - \beta F_{AA} + \gamma F_{DD}} \quad (1)$$

where $F_{DD}$ is the observed fluorescence intensity in the donor (green) channel, $F_{DA}$ is the observed fluorescence intensity in the acceptor (red) channel through indirect excitation and $F_{AA}$ is the observed fluorescence intensity in the acceptor (red) channel through direct excitation (PIE pulse).

## Liquid-to-solid phase transition (LSPT) puncta isolation and ThT staining

One hundred microliters of an LLPS reaction containing 25 μM of unlabeled, monomeric Tau441 with or without 25 μM $\alpha$S in LLPS buffer (supplemented as explained above) were spotted onto non-Binding 96-Well Microplates, covered with an adhesive foil, and droplet formation was verified after 10 min of equilibration by WF microscopy. After 48 h of incubation at room temperature, the presence of protein rafts and puncta was confirmed. Then, the liquid on top of the rafts was carefully removed from the well, and then 50 μL of isolation buffer (10 mM HEPES, pH 7.4, 1 M NaCl, 1 mM DTT) were added and incubated for 10 min. The high salt concentration ensured that no LLPS could reoccur due to residual PEG and that possible protein assemblies formed merely by electrostatically-driven interactions would be disassembled. The bottom of the well was then gently scraped using a micropipette

tip and the resulting solution was transferred onto an empty well for visualization. The presence of isolated puncta was verified by WF microscopy after incubating the sample for 1 h with 50 µM ThT. Sonicated αS fibrils were prepared by incubating 300 µL of a 70 µM αS solution in PBS pH 7.4, sodium azide 0.01% for 7 days at 37 °C and 200 rpm in an orbital shaker. Then, the solution was centrifuged for 30 min at 9600 × g, the pellet was resuspended in PBS pH 7.4 and sonicated (1 min, 50% cycles, 80% amplitude in a Vibra-Cell VC130 Ultrasonic Processor, Sonics, Newton, USA) to generate fibrillar samples with a relatively homogeneous size distribution of small fibrils.

### Single-molecule fluorescence analysis of the isolated LLPS-derived αS/Tau aggregates

FCS/FCCS and two-color coincidence detection (TCCD) analysis were performed in the same MT200 time-resolved fluorescence confocal microscope (Pico-Quant, Berlin, Germany) as the FLIM-FRET miscroscopy experiments using the PIE mode. The laser powers for these experiments were adjunted to 6.0 µW (481 nm) and 6.2 µW (637 nm). The combination of these laser powers were chosen to obtain similar brightness for the fluorescence pair used, at the same time that optimal count rates were obtained and photobleaching and saturation were avoided. Both data acquisition and analysis were performed on the commercially available software SymphoTime64 version 2.3 (PicoQuant, Berlin, Germany).

The LLPS-derived isolated αS/Tau aggregates samples were diluted in isolation buffer to a suitable single-molecule concentration (typically a 1:500 dilution, since the aggregates are already at low concentration when isolated from the coacervate sample). The sample was spotted directly onto a cover glass (Corning, Corning, USA) previously coated with a 1 mg/mL BSA solution.

For the PIE-smFRET analysis, a lower intensity threshold of 25 photons in the green and red channel was applied to filter out the low-intensity signal arising from the monomeric events (note that monomers are in excess with respect to the aggregates in the isolated aggregate sample). This threshold was calculated as five times the mean intensity of monomeric αS obtained from the analysis of a sample of pure monomers in order to exclusively select the aggregates for the analysis. The PIE excitation scheme together with the TSCPC acquisition enabled the application of a lifetime-weighted filter, which aided removal of background and spectral cross-talk. The intensities of the burst selected using the abovementioned threshold were corrected by the average background signal, which was determined by the occurrence vs intensity/bin histogram of a sample containing only the buffer. The bursts associated with big aggregates typically occupy more than one consecutive bin (set to 1 ms) in the time traces. In these cases, the maximum intensity bin was selected. For the FRET and stoichiometry analysis, a theoretically-determined gamma factor, $\gamma$, was used (0.517). With the excitation laser powers used, the spectral cross-talk and direct excitation contributions are negligible (experimentally determined). Burst-wise FRET efficiency and stoichiometry were calculated as given by

$$E = \frac{F_{DA}}{F_{DA} + \gamma F_{DD}} \tag{2}$$

$$S = \frac{F_{DA} + \gamma F_{DD}}{F_{DA} + \gamma F_{DD} + F_{AA}} \tag{3}$$

where $F_{DD}$ is the fluorescence intensity in the donor (green) channel, $F_{DA}$ is the fluorescence intensity in the acceptor (red) channel through indirect excitation and $F_{AA}$ is the fluorescence intensity in the acceptor (red) channel after direct excitation by PIE pulse. To estimate the number of Atto647N-Tau and AF488- αS, the intensities for each burst/aggregate was divided by the average intensity of the monomeric proteins, which were determined in independent experiments. The

experiments were performed in triplicate and measurements of 30–60 min were performed in order to detect a statistically-relevant number of aggregates per sample.

The same intensity time traces were also analyzed by FCS/FCCS. The effective focal volume of the green channel ($V_{eff,g}$) and its structural parameter ($\kappa$) in our system were determined using a 1 nM solution of Atto488 (ATTO-TEC GmbH, Siegen, Germany) yielding $V_{eff,g} = 0.36$ fL and $\kappa_g = 2.71$. A dual-labeled dual-stranded DNA (dsDNA) with AF488 and Atto647N was used for the determination of the red and dual-color effective focal volume and their structural parameters, yielding $V_{eff,r} = 0.27$ fL, $V_{eff,gr} = 0.22$ fL, $\kappa_r = 2.76$, and $\kappa_{gr} = 2.93$, respectively. Positive and negative cross-correlation controls were performed with the dual-labeled dsDNA before and after heat denaturation. Auto-correlation (AC) curves for both the green and red channel were typically fitted with a 2 diffusion-component model accounting for residual monomeric protein in the samples, while the cross-correlation (CC) curves were fitted with a 1-component simple diffusion model[44].

For measuring the diffusion coefficient of αS in the diluted phase in the different LLPS systems (Fig. 3c, Supplementary Fig. 3c), 50 µL of a freshly filtered (0.22 µm syringe filter) solution containing 5 nM AF488-labeled αS and 100 µM unlabeled αS in LLPS buffer (HEPES 10 mM pH 7.4, 100 mM NaCl, 15% PEG-8) were spotted onto a µ-Slide Angiogenesis well plate. Intensity time traces for a triplicate experiment were acquired for 1 min focusing the laser 20 µm above the well surface for an optimal objective lens working distance. Auto-correlation curves ($G(t)$) were fitted with a 1 diffusion-component model with a blinking term accounting for the triplet state of the dye using the following equation:

$$G_i(\tau) = G_i^0 \left(1 + \frac{T e^{-\frac{\tau}{\tau_T}}}{1 - T}\right) \frac{1}{\left(1 + \frac{\tau}{\tau_{D,i}}\right)\sqrt{1 + \frac{\tau}{\kappa^2 \tau_{D,i}}}}, \tag{4}$$

where $G(t)$ is the correlation amplitude, $i$ denotes either the green or the red channel, $T$ denotes the amplitude of the triplet state, $\tau_T$ is the blinking relaxation time, $\tau$ is the correlation time, $\tau_D$ is the diffusion coefficient and $\kappa$ is the structure parameter of the corresponding focal volume.

### Statistical analysis

For the FLIM and ThT imaging statistical analysis, significant differences were assessed with two-sample, two-tailed t-tests assuming unequal variances after finding significant differences between variances via Levene's tests using Excel (Excel 2000 9.0.3821 SR-1). Significance was set at $p < 0.05$. All data were plotted using OriginPro 9.1. The exact $p$ values for all the analysis performed in this study are reported in Supplementary Table 1.

### Reporting summary

Further information on research design is available in the Nature Research Reporting Summary linked to this article.

## Data availability

The data generated in this study are provided in the Source Data File. Source data are provided with this paper.

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

## Acknowledgements

This work has been funded by MCIN/AEI/ 10.13039/501100011033 and "ERDF A way of making Europe", by the "European Union" (grants: PGC2018-096335-B100 to N.C., PID2019-109276RA-I00 to J.O. and PID2019-109306RB-I00 to D.V.L.) and the Centro Universitario de la Defensa de Zaragoza-University of Zaragoza (grants: UZCUD2019-BIO-01 and UZCUD2020-BIO-01 to N.C. and I.G.), and is also part of a project that has received funding from the European Union's Horizon 2020 research and innovation program under the Marie Skłodowska-Curie grant agreement No. 813209 (I.G.). J.O. is a Ramón y Cajal Fellow (grant RYC2018-026042-I funded by MCIN/AEI/10.13039/501100011033 and by "ESF Investing in your future"). NMR experiments were performed in the "Manuel Rico" NMR Laboratory (LMR) of the Spanish National Research Council (CSIC), a node of the Spanish Large-Scale National Facility (ICTS R-LRB). Authors would like to acknowledge Prof. Alberto Boffi for kindly providing highly pure recombinant K18 protein, the use of Servicio General de Apoyo a la Investigación-SAI and the National Facility ELECMI ICTS, node "Laboratorio de Microscopias Avanzadas" at Universidad de Zaragoza and Servicios Científico Técnicos del CIBA (IACS-Universidad de Zaragoza).

## Author contributions

N.C. conceived the project. P.G. carried out and analyzed the FRAP., FCS. and FRET. experiments. P.G., D.P. and N.C. performed the FLIM imaging and smFRET and FCS/FCCS analysis of the isolated aggregates. I.S., M.B. and I.G. performed and analyzed the EPR experiments, while J.O. and D.V.L. did the NMR experiments. P.G., D.P. and J.T. produced all the protein variants and performed the rest of the experiments. P.G. and N.C. prepared the paper with contributions from all the authors.

## Competing interests

All authors declare no competing interests.
