## [Peer Review File · Nature Communications]

REVIEWER COMMENTS

Reviewer #1 (Remarks to the Author):

Gracia et al. show that alpha-synuclein undergoes complex coacervation with poly-Lys and Tau. The authors apply a variety of techniques, including NMR, EPR, FLIM and FRET to characterize interactions between alpha-synuclein and Tau in LLPS conditions in vitro in the presence of crowding agents. By NMR, EPR and the use of C-terminal truncated alpha-synuclein mutant, the authors showed that interactions are driven by electrostatic interactions. The following are this reviewer's questions/comments/suggestions:

1. Can phase separation still be observed for alpha-synuclein and/or Tau without the use of crowding agents, which by themselves are known to phase separate in water-salt systems and thus potentially making data interpretations less straightforward? For example, is the inability of 1,6-hexanediol to perturb the alpha-synuclein/pLK droplets due to potential PEG-hexanediol interaction/non-interaction?

Hey et al. 2005 Polymer, The salting-out effect and phase separation in aqueous solutions of electrolytes and poly(ethylene glycol)

Kaul et al. 1995 Biotechnol. Bioeng., Kinetics of Phase Separation for Polyethylene Glycol-Phosphate Two-Phase Systems

2. Although approximate conditions are described in the main text (p.3 lines 116-7), include specific solution conditions in the figure legends for clarity. For example, it is unclear whether the Fig. 1C results were performed in 0 vs. 500 mM or 100 vs. 600 mM, which could make a difference in data interpretation. (minor)

3. The link to amyloid aggregation as presented in Fig.5 is not convincing as the ThT-positive signals observed are localized, with minimal sign of propagation as expected for amyloids. Increasing the incubation/maturation time could be useful to let the aggregation process evolve. In addition, transmission electron microscopy can also be used to definitively demonstrate the presence of amyloid fibrils.

4. Similar to #3, the FLIM data shown in Fig. 6 is inadequate as proof for liquid-to-solid phase transition as change in fluorophore fluorescence lifetime is indicative only of change in probe environment. Again, electron microscopy can give a more direct proof for the presence of fibrils (depicted as amyloid co-aggregates in Fig. 7).

5. For Figs. 6 and Suppl. Fig. 9, the color scales is best kept constant and chosen to maximize color contrast – it's difficult to see cyan on a background of green. (minor)

6. For the FRET data presented in Suppl. Fig. 10, because of the high initial concentrations of labelled proteins (i.e., 1 micromolar each of AF488-labelled alpha-synuclein and Atto647N-labelled Tau variant), which translates to even higher concentrations in the condensates, changes in donor lifetimes can not only be due to intermolecular FRET but also, if not dominated by, fluorescence quenching. Given this, can the observed changes in calculated FRET efficiencies be interpreted as being mainly due to changes in protein conformation?

Reviewer #2 (Remarks to the Author):

In their manuscript titled 'Molecular Mechanism for the Synchronized Electrostatic Coacervation and Amyloid Co-Aggregation of Alpha-Synuclein and Tau, Gracia et al characterize the co-condensation of recombinant alpha-Synuclein and Tau in vitro. They use a set of sophisticated biophysical methods to describe the protein conformations of aSyn and Tau in condensates.

They find that negatively charged aSyn coacervates with Tau in crowding conditions, that the produced condensates retain liquidity for a long time, and that over time Tau aggregates of Tau form in solutions with aSyn, Tau, and PEG.

Although the techniques used are of interest for the LLPS community, the findings bare little new information: it is known that both Tau and qSyn independently and together form condensates that transition into aggregates, and it has been shown that Tau coacervates with a number of polyanionic polymers, such as aSyn. In summary, the data seem incomplete, provide not enough novelty, and do not support the conclusion.

Major concerns:

The most problematic major confound in the study I see in the extremely high % of PEG that the authors use to induce Tau-aSyn co-condensation. Cellular crowding creates a colloid pressure comparable to 1-2% PEG, whereas the authors use 15% PEG. In these conditions, Tau will co-condensates with almost every negatively charged polymer.

Another major confound is that there are no data presented on aSyn and Tau alone, especially for the liquid-to-solid-phase-transition experiments. Reading other literature on Tau condensation, it seems that aSyn in fact may inhibit aggregation of Tau rather than promoting it, as suggested by the title.

Furthermore, no evidence of 'amyloid co-aggregation' is given, but it seems that amorphous protein aggregates of Tau forming in the highly crowded solution in fact exclude aSyn (see FLIM images).

Also, Thioflavine fluorescence intensity increases upon inhibition of its free rotation in solution, and therefor usually also indicates an increase in viscosity in condensates, and it does not only label amyloid structures. Dot like Thioflavine+ structures cannot be taken as a sign of amyloid aggregation (they may resemble any aggregation or sticking of the Thioflavine) if no comparison to conditions with lower Thioflavine signal is shown for comparison. Especially for the distinction of liquid condensates and aggregates, the use of Thioflavine fluorescence is highly controversial since both conditions largely increase Thioflavine fluorescence.

Reviewer #3 (Remarks to the Author):

The article of Gracia P. et al entitled "Molecular mechanism for the synchronized electrostatic coacervation and amyloid co-aggregation of alpha-synuclein and Tau" addresses a very interesting topic in the context of understanding the formation of amyloid aggregation from protein condensates. The paper is very well written in a clear progressive way that allows the reader to understand the different steps containing a lot of data (and supplementary data) from different biophysical techniques. The results are clearly presented and discussed and all the figures are very clear and convincing. The work is focused on the behavior of alpha-synuclein (alphaS) when it phase separates into liquid condensates with different positively charged polypeptide: poly-L-lysine and the protein Tau either full length or truncated from its N-ter part. They demonstrated the electrostatic nature of the coacervates and that alphaS plays a scaffolding role. They observed the formation of amyloid co-aggregation (alphaS/Tau) inside the liquid coacervates and determined the key factors leading to the formation of the aggregates. All the results brought by this study led to the proposition of a model for the synchronized formation of either liquid-like or gel-like structures. This mechanism has a potential relevance in the context neurodegenerative diseases involving amyloid formation of alphaS.

As my field of expertise is EPR spectroscopy and in particular SDSL-EPR, my main questions are on this part of the article. The results obtained from the labeling of two Cys variants of alphaS showed very little or no spectral modification resulting from the phase separation leading to the information that alphaS keeps a high degree of flexibility.

1. Could the authors justify the choice of their nitroxide labels ? Why did they choose TEMPOL rather than the most currently used MSTL ? Maybe spectral variations could be better detected using MTSL thanks to its higher flexibility compared to maleimido-TEMPOL.
2. The authors excluded the effect of viscosity on the (slight) spectral shape modification at position 122. I was thinking on another potential effect of broadening that could come from spin-spin interaction between labeled proteins in the droplets characterized by a high protein concentration. Could the authors reject this possibility ?
3. Line 252-255, the authors concluded that NMR experiments corroborates EPR results. My point is that by NMR they only gained information from the dispersed phase (due to sedimentation problem) whereas EPR results come from both dispersed and condensed phases. This is not clear to me how they can compare results from samples with different contents.
4. In the discussion section (L359), the authors claimed that they worked at low micromolar concentrations and physiological relevant conditions. To my point of view 25-100 μ M protein concentration is still high and I am not sure that it reflects physiological conditions. Could the authors comment what they mean by “physiological relevant conditions” ?

Minor points and typos

Line 476: “degree of labelling was confirmed ...” Could the authors give more quantitative information % of labeling ?

Line 569 and 570: sub- or upper-scripts are missing for D2O, 1H and 15N

Line 583: “ μ L” instead of “uL”

Legend fig2: I was confused by the different indications concerning the scale bars of the images. Please check.

Legend fig3: Band II and band III need to be defined or even indicated in the figure.

Sup Fig5 legend. The authors simulated their EPR spectra with a two-component isotropic model. It would be interesting to have the results of the simulation in terms of rotational correlation time and % of population (besides giso and A-values).

Reviewer #4 (Remarks to the Author):

The manuscript by Gracia et al uses a range of experimental approaches to characterize and in part quantify the coacervation between alpha-Synuclein and Tau. The manuscript is overall relatively clearly written and probes a potentially interesting and perhaps (biologically) important interaction. This

biological relevance is, however, still somewhat unclear. My major concern with the work is that while many experiments are performed, a number of them are rather qualitative. This makes it difficult to understand what is going on, and the authors use words such as client/driver in a multi-component system w.o. really characterizing the interactions thermodynamically. In addition, in places I find the language overly strong compared to what is actually shown. Overall, the paper probes a potentially interesting interaction which could be important. But in the end, it is somewhat unclear exactly what the authors show. I would recommend that the authors focus their message and tone down claims that are not supported by quantitative measurements and analyses.

Major

1.

Already in the abstract the authors write “leading to the co-aggregation of both proteins inside the coacervates through a mechanism that could be potentially highly relevant in disease.”. First, the term co-aggregation is somewhat vague and from reading the paper it is still not clear to me whether the authors really show that the two proteins form aggregates “together” (as in with some kind of semi-specific interactions between the two proteins) or whether the experimental conditions simply result in aggregates of both proteins. Also, the statement “that could be potentially highly relevant in disease” is simultaneously vague (“could be potentially”) and strong (“highly relevant”), and I don’t see any evidence from this paper or the discussion that this interaction is relevant for disease. It could well be and obviously both proteins are important. But I think the results do not say much in terms of disease mechanism or progression and would urge the authors to tone down such a link. If not, then at least clarify what they have shown rather than the vague statement at the end of the abstract.

2.

I think in places that the authors are underplaying the previous results from Zweckstetter et al (ref 36). For example, on p. 2 they write “To the best of our knowledge, the only polymers reported so far to undergo complex coacervation with Tau are RNA molecules or heparin²¹”. While it is true that in Ref 36 aSYN was only shown to partition into condensates formed by Tau and RNA, these authors did show condensates with Tau/RNA/Synuclein. I think it would be important that the authors make it clearer what was previously done.

Later (p. 3) the authors write “Owing to this interaction, τ S has been recently reported to exhibit a client role and partition into preformed Tau/RNA electrostatic coacervates³⁶”, which is a better description. But in the end, it is about the thermodynamics and interactions and how much the three components and their concentrations shift the phase diagrams, and hence the word “client” is somewhat vague (see also below) and somehow downplays the previous work which I don’t think is fair.

In the work by Zweckstetter et al, the authors (I think) obtained condensates of Tau(50 μ M)/polyU(60nM)/aSYN (10 μ M) and 2.5% dextran (and in some experiments 10% dextran). In the current work, phase separation appears to occur at 5–10 μ M Tau, 50 μ M aSYN and 15% PEG (or 20% dextran).

Thus, while it is interesting that the current work shows that one can get PS of aSYN/Tau in the absence of polyU, it appears that they instead need to drive PS by addition of greater concentrations of crowding agent. While that is OK (as long as it is made clear), I do think it is important to make it clearer that perhaps the two sets of basic observations are not so different. Instead, the differences (in my opinion) come from the additional characterization of the condensates here.

3.

On p. 3 wording like “co-driving model” is in my opinion vague. As above, it’s likely just about thermodynamics and whether Tau alone, aSYN alone or the two together can phase separate likely “just” depends on the concentrations and the conditions (presence of crowder, poly-U, salt, etc). Unless the authors quantify the thermodynamics via measurements of phase diagrams (beyond the data in Fig. 2c), I don’t think they have evidence to really say what is driving and what isn’t. This is also important on p. 4 where the authors discuss an “active role” vs “client”. Again, such statements would be much clearer to understand if the authors quantified the phase properties/diagrams. I would suggest they either do that or remove claims about what is driving and what isn’t.

Similarly, what do the authors really mean on p. 9 by the statement “Here, we prove that aS is indeed capable of co-driving the LLPS process with Tau through the same type of interactions but with an active role rather than being a simple client molecule.” What do they mean by “co-driving” and “active”?

4.

On p. 3: I really don’t think it is fair to say that 5-15% PEG is a small amount, in particular since most experiments are done with 15% PEG. Indeed, I think that the authors should make it clearer that—at least with the protein concentrations used—they need relatively high concentrations of PEG or dextran to the two proteins to PS. Previous experiments on PEG on IDPs in dilute solution (e.g. A. Sorzano et al, PNAS, 2014) show relatively strong effects. Even better of course, they would quantify the phase behaviour via measuring phase diagrams at higher protein concentrations in the absence or lower amounts of PEG. In the absence of such measurements, they should at least make it clearer how they think the interaction would be physiologically relevant.

5.

On p.4 the authors argue that the complex coacervation between aSYN and Tau is electrostatically driven. I would like the authors to explain more clearly how they show this. First, addition of salt does not just affect ionic interactions. Second, the Tau condensates alone can also be dissolved via addition of NaCl, so the salt dependency of the aSYN/Tau condensates would likely be affected by salt even if the interactions between aSYN and Tau were not ionic. I realize that the salt dependencies are different, which likely has some evidence for the interaction, but the authors do not really discuss or quantify this. It would be good with a more quantitative argument for the nature of the interaction.

6.

On p. 6 a change in translational diffusion of 2x is stated to be “almost identical” and a change in 13x is listed as “slightly faster”. Obviously, these differences are much smaller than the 50–600 fold differences to the dilute phase (these are stated to be “more than 2 orders of magnitude”, but that’s only relative to Tau condensates, not with pLK). I would urge the authors to make the wording somewhat more neutral rather than saying that the 50–600 fold are remarkable (why?) and the 2–13 fold are “almost identical”/“slightly faster”.

7.

In the EPR experiments described on p. 6 I lack a description of how much of the protein that is probed is in condensates and how much is in a dilute phase. What do these experiments say about the interactions within the condensates?

8.

On p. 7 the authors write:

“When we analyzed the structure and dynamics of the protein that remains in the dispersed phase in the LLPS samples by NMR (Supp. Fig. 5c and d), we observed that the protein behaves almost identically in the presence of pLK and Δ Nt-Tau, both in terms of secondary structure and protein backbone dynamics, as detected by secondary chemical shifts and R1rho relaxation experiments. The NMR data corroborate the EPR results, showing that the C-terminus of Δ S undergoes the main loss of conformational flexibility, while maintaining the disordered nature as the rest of the protein sequence, under LLPS conditions with poly-cations.”

Given that these experiments report on the protein in the dilute phase, it is unclear in which way they “corroborate the EPR results” (and which EPR results). Also, the sentence “while maintaining the disordered nature as the rest of the protein sequence, under LLPS conditions with poly-cations” it should be clarified that the protein is the same in the dilute phase, and that the experiments do not say anything about the condensates. Similarly, the SI Fig. S5 legend says “LLPS context” which again might be made clearer by saying the dilute phase under conditions where the proteins can phase separate.

9.

On p. 7/8/9, I would like the authors more clearly to explain how they think about the co-aggregates of aSYN and Tau. They in places seem to imply that there are direct interactions (FRET; although the values are low) and that they are amyloid (ThT), but it is unclear whether they think they are forming amyloids with both proteins in a single fibril/protofibril or what is going on. How should I interpret the FRET data in Fig. S10 at a molecular level?

Minor

Abstract: In my opinion, there is no reason to call the methods as advanced biophysical techniques and I would delete “advanced”.

p. 2: Phase separation does not have to occur through weak interactions.

p. 2: In addition to the examples of FUS and TDP-43, the authors might add hnRNPA1 (and perhaps hnRNPD) as proteins that have been shown to form aggregates and to phase separate, and where there (at least for hnRNPA1) is a link between the two.

p. 2: In “One of these proteins is Tau”, it is unclear what “these” refers to. Please clarify.

p. 2: In “Tau has been shown to trigger LLPS” it is unclear what is meant by “trigger”. (Same on the use of the word on p. 9)

p. 3: The authors use pLK of “100 residues”. How pure is this (i.e. what is the range in degree of polymerization)?

p. 3: “First, we confirmed the electrostatic interaction of pLK with the Ct-domain of τ S by solution NMR spectroscopy (Fig. 1b)”. While I do not doubt that the interaction between pLK and aSYN has a strong electrostatic component, the NMR experiments do not show it is electrostatic.

p. 4: “we found that both crowding agents distribute homogeneously over the whole sample without showing a segregative nor an associative behavior”. Can the authors quantify the relative concentrations in dilute and condensed phases to support this?

p. 7: “Given such differences in affinities between aS and the various poly-cations, we hypothesized that their liquid properties might evolve differently over time.” Could the authors unpack this argument and how the differences in affinity would lead to differences in time-dependent changes.

p. 7: What is “raft”-like about the “protein rafts”? Why this name? By analogy with lipid rafts?

p. 10: How do the authors know the droplets have a low surface energy? And what do they really mean?

p. 11: On p. 11 what do the authors mean by “coacervate valence”? And in general, what do they really mean by “valence” and “valence exhaustion” in the discussion, and which data reports on this?

p. 14: μM -> μM , and sub/superscripts missing from D2O, ^1H and ^{15}N

p. S6: In Supporting Fig. 5a, it would perhaps be easier to see what is going on if the data were referenced to (subtracted by) the chemical shift differences in free aSYN.

REVIEWER COMMENTS

Reviewer #1 (Remarks to the Author):

Gracia et al. show that alpha-synuclein undergoes complex coacervation with poly-Lys and Tau. The authors apply a variety of techniques, including NMR, EPR, FLIM and FRET to characterize interactions between alpha-synuclein and Tau in LLPS conditions in vitro in the presence of crowding agents. By NMR, EPR and the use of C-terminal truncated alpha-synuclein mutant, the authors showed that interactions are driven by electrostatic interactions. The following are this reviewer's questions/comments/suggestions:

1. Can phase separation still be observed for alpha-synuclein and/or Tau without the use of crowding agents, which by themselves are known to phase separate in water-salt systems and thus potentially making data interpretations less straightforward? For example, is the inability of 1,6-hexanediol to perturb the alpha-synuclein/pLK droplets due to potential PEG-hexanediol interaction/non-interaction?

We agree with the reviewer that caution needs to be taken when using macromolecular crowders such as PEG, ficoll or dextran in such experiments to avoid undesirable effects. For this reason, we have carefully analyzed the two crowders used (PEG and dextran) and discarded any segregative or associative effects of the crowders in the LLPS systems (see Supp. Fig 1). Also, we have shown that two structurally different crowders have similar behaviours in our systems, which is consistent with a general common effect related to the steric repulsion phenomenon as the main contributor of the crowders in facilitating LLPS in our study. The typical conditions that we have used consist of physiologically relevant salt concentrations (100 mM NaCl) and moderate crowder concentrations (15% PEG), although lower crowder concentrations can be used to visualize phase separation.

We have included in the revised version of the manuscript a panel in Supp. Fig. 2, where we show phase separation of alpha-synuclein and the N-terminally truncated form of Tau in the presence of 5 % PEG with and without 1,6-hexanediol: protein droplets are formed under the two conditions without significant differences.

2. Although approximate conditions are described in the main text (p.3 lines 116-7), include specific solution conditions in the figure legends for clarity. For example, it is unclear whether the Fig. 1C results were performed in 0 vs. 500 mM or 100 vs. 600 mM, which could make a difference in data interpretation. (minor)

We have now added the specific solution conditions to the figure legends for additional clarity.

3. The link to amyloid aggregation as presented in Fig.5 is not convincing as the ThT-positive signals observed are localized, with minimal sign of propagation as expected for amyloids. Increasing the incubation/maturation time could be useful to let the aggregation process evolve. In addition, transmission electron microscopy can also be used to definitively demonstrate the presence of amyloid fibrils.

The data presented in Figure 5 corresponded to the initial evidence showing that the punctual aggregates formed inside the liquid coacervates were able to bind ThT, as the amyloid aggregates do. In order to further investigate this idea, we isolated these aggregates from the

liquid coacervates and obtained a dispersed solution of aggregates (previous Supp. Fig. 8). With these samples of isolated aggregates, we tested the ability of these aggregates to bind ThT and increase its fluorescence quantum yield in the typical in vitro ThT experiments. These are well-established assays to check the amyloid nature of dispersed aggregates in a solution. We found that the LLPS-derived isolated aggregates were able to bind and increase the fluorescence quantum yield of ThT similar to the canonical amyloid fibrils do. Also we showed that these aggregates are resistant to high salt concentrations, indicating that non-electrostatic interactions contribute to the stability of these aggregates. By contrast, they are sensitive to high concentrations of chemical denaturant agents; this behaviour is typical of amyloid-like aggregates.

In order to further prove the amyloid-like nature of these aggregates we have followed the reviewer's proposal and performed imaging analysis of the dispersed solution of the aggregates. We used atomic force microscopy (AFM) of the isolated LLPS-derived α S/Tau441 aggregates, as this imaging technique provides information similar to that obtained by negative staining transmission electron microscopy and typically more resolution in the z-axis. A representative AFM image of these aggregates, and in comparison to the typical α S amyloid fibrils, can be now seen in Supp. Fig. 10 of the revised version of the manuscript. The aggregates as visualized by AFM do not show the typical fibrillar morphology, which can be explained by the highly crowded nature of the environment in which they have grown, which would restrict their growth and propagation. Other amyloid-like aggregates of alpha-synuclein generated under highly crowded conditions were reported to show a globular rather than fibrillar morphology (Camino J.D. et al., *Chem. Sci.* 2020, 11, 11902-14). We have also observed that the LLPS-derived α S/Tau441 aggregates tend to associate under the high salt concentrations that we use to isolate the aggregates. This is typically observed for amyloid fibrils and is a characteristic of a high hydrophobic surface area in the aggregates (note that amorphous aggregates tend to aggregate burring the hydrophobic surface exposed areas). Other non-fibrillar aggregates with globular morphology have also been reported to acquire an amyloid-like structure (Tau: Nie C.L. et al., *Plos One* 2007, 2, e629; α S: Camino J.D. et al., *Chem. Sci.* 2020, 11, 11902-14; A β peptide: Bhatia R et al., *FASEB J.* 2000, 14, 1233-43; PrP: Sanghera N et al., *Biochim. Biophys. Acta* 2008, 1784, 873-81), in some cases demonstrating that they present a cross-beta structure despite their being inefficient at elongating (Camino J.D. et al., *Chem. Sci.* 2020, 11, 11902-14). In addition, the LLPS-derived α S/Tau441 aggregates are mostly hetero-aggregates (as we have determined -see the new single-molecule fluorescence experiments), which could lead to a different ultrastructure as compared to the homo-aggregates.

The concentrations of the isolated aggregates in the samples are, however, very low (in the low nM range), and attempts to concentrate them using centrifugation or filter devices have failed. The excess of residual monomeric protein in the solution also hampers average structural characterizations. Nevertheless, we have now performed new single-molecule fluorescence experiments of the isolated aggregates and determined the fraction of hetero-aggregates composed of α S and Tau. Most of the aggregates generated inside the coacervates were hetero-aggregates composed of both α S and Tau (ca. 60 % of the aggregates contain both α S and Tau), although a preference for Tau protein in the co-aggregates was observed, in agreement with the FLIM results. Also, ca. 10% of the aggregates contained only α S, and ca. 30% only Tau. These experiments have provided relevant information on the composition of these aggregates and we have included them in the revised version of the manuscript (see new Fig. 6) - see figure below:

Isolated LLPS-derived α S/Tau aggregates, 24h

Figure 1. Single-molecule fluorescence analysis of the aggregates generated inside α S/Tau coacervates after being isolated from the LLPS solutions. **a)** A representative single-molecule fluorescence time trace with a burst corresponding to an α S/Tau441 co-aggregate (burst above the assigned thresholds) is shown for the three detection channels (emission of AF488 and Atto647N after direct excitation, blue line and red lines, respectively, and emission of Atto647N after indirect excitation, FRET, violet line). **b)** FCS/FCCS analysis of a sample of isolated α S/Tau441 aggregates generated inside the coacervates (left panel). The auto-correlation (AC) curves for AF488 and Atto647N are shown in blue and red, respectively, and the cross-correlation (CC) curve, related to the aggregates containing both dyes, is shown in violet. The AC curves reflect the presence of both labeled monomeric and aggregated protein species, while the CC curve only shows the diffusion of dual-labeled aggregates. The same analysis but with a sample containing only monomeric α S and Tau441 under the same solution conditions as the isolated puncta is shown in the right panel as control. No CC curve is observed. **c)** Single-molecule fluorescence burst analysis of isolated aggregates generated inside α S/Tau441 electrostatic coacervates. The information of each aggregate detected in four different replicas ($N=152$) was plotted according to their stoichiometry, S , and FRET efficiency value (upper panel, the color scale reflects occurrence). Three types of aggregates can be distinguished: only- α S aggregates, with $S \sim 1$, $FRET \sim 0$, only-Tau aggregates, with $S \sim 0$, $FRET \sim 1$, and Tau/ α S hetero-aggregates, with intermediate S and FRET values. An estimation of the number of the two labeled-proteins in each hetero-aggregate detected ($N=100$) is shown in the bottom panel (the color scale reflects occurrence). The hetero-aggregates are typically enriched in Tau protein, in agreement with the FLIM data.

In order to corroborate further the amyloid nature of the aggregates generated in the interior of the α S/Tau441 liquid coacervates, we have performed new experiments with a well-established Tau variant that has been demonstrated to be unable to form amyloid aggregates. In this variant, the main regions responsible for Tau amyloid aggregation, residues 275-311, have been removed (von Bergen M et al., *BBA* 2005, 1739, 158-166; Mukrasch M et al., *JBC* 2005, 280,

24978-86; Zibae S et al., *Prot. Sci.* 2007, 16, 906-18; Seidler P.M. et al., *Nat. Chemistry* 2018, 10, 170-176). We have generated this variant, that we have referred to as Aggregation Deficient Tau variant, or AggDef-Tau, and reproduced previous results on the inability of this variant to initiate amyloid self-assembly in the typical *in vitro* experiments for Tau amyloid formation (see new Supp. Fig. 11 a,b). We have also observed that the LLPS properties of Tau are not altered in this variant, as expected since the deleted region has not major contributions in this process, and, thus, it forms liquid droplets and rafts with α S with identical properties to those of the full-length Tau protein (see new Supp. Fig. 11 c,d). However, we could not detect the formation of puncta by FLIM and when performing the aggregate isolation protocol used for isolating α S/Tau441 aggregates from the liquid coacervates, only few aggregates were recovered (almost 10-times less than with Tau441), which also mainly contained α S, in contrast to the α S/Tau441 aggregates (see new Supp. Fig. 11 c,d). These results indicate that the amyloid essential region of Tau protein is also essential for the formation of α S/Tau aggregates inside the liquid coacervates, further demonstrating the amyloid nature of these aggregates. All these new results, which are copied below to facilitate the review process, are incorporated in the revised version of the manuscript (Supp. Fig. 11).

Figure 2. LLPS and LSPT of αS and an amyloid aggregation deficient Tau variant. a) Schematic of the different protein regions of Tau and the amyloid aggregation deficient Tau variant, AggDef-Tau, where the region of full-length Tau between residues 275 and 311 was removed. b) Aggregation kinetics of Tau441, K18 and the AggDef-Tau variants under typical in vitro Tau amyloid aggregation conditions (50 μM of each Tau variant in PBS buffer, 0.02% azide, in the presence of 12.5 μM heparin and 25 μM ThT). c) Brightfield (BF, left image) and widefield fluorescence (WF, center and right images) microscopy images of αS /AggDef-Tau coacervates (50 μM each protein, 1 μM AF488-labeled αS and Atto647N-labeled Tau variant, in LLPS buffer) at 0 h of incubation. d) FLIM images of analogous coacervate samples at 5 h of incubation (suspended droplets in the left panel, a raft covering the image in the right panel). The lifetime shown corresponds to Atto647N-AggDef-Tau. e) Single-molecule fluorescence burst analysis of isolated aggregates generated inside αS /AggDef-Tau electrostatic coacervates. Two representative single-molecule fluorescence time traces are shown at the top, where the bursts of aggregates are evident as the only bursts whose intensities are above the established

thresholds. These bursts are absent in monomer-only samples (not shown). An example of FCS/FCCS analysis of the isolated aggregates generated inside the α S/AggDef-Tau coacervates is shown at the bottom left panel, where very few aggregates were detected (almost no cross-correlation curve, CC in violet, and the auto-correlation curves, AC in blue and red, showed mainly particles diffusing as monomers). The stoichiometry, S, vs FRET efficiency plot of the few aggregates detected is shown at the bottom-center panel (the color scale reflects occurrence). Most of the aggregates detected contained mainly α S molecules. For the very few hetero-aggregates detected, the estimated number of Atto647N-AggDef-Tau and AF488- α S molecules per aggregate is shown in the bottom-right panel (the color scale reflects occurrence).

We understand the concerns raised by the reviewers about the amyloid nature of the aggregates from the experimental evidences showed in the previous version of the manuscript. We believe that we have now provided additional evidences that indicate that the aggregates generated inside the α S/Tau441 condensates have an amyloid-like nature and made an effort to clarify previous evidences that seemed to not be clear enough through the text. We have also considered removing the word “amyloid” from the title, although we believe that we have enough experimental evidences to describe them as amyloid-like aggregates throughout the text.

4. Similar to #3, the FLIM data shown in Fig. 6 is inadequate as proof for liquid-to-solid phase transition as change in fluorophore fluorescence lifetime is indicative only of change in probe environment. Again, electron microscopy can give a more direct proof for the presence of fibrils (depicted as amyloid co-aggregates in Fig. 7).

In the FLIM images, we observed that there is an increase in intensity, particularly in Atto647N-Tau, indicative of an increase in fluorophore density, concomitant with a reduction in the lifetime of the fluorophore (Supp. Fig. 8a), which suggests that the decrease in fluorescence lifetime is related to an increase in quenching effects due to the proximity of fluorophore molecules. The solid nature of the puncta is proven by their resistance to high salt concentrations and we have now shown direct evidences of their existence by AFM and single-molecule fluorescence (see new Supp. Fig. 10). It is also possible that changes in probe microenvironment could contribute to the reduction in fluorescence lifetime, but in any case, our results indicate that the formation of aggregates inside the liquid coacervates is associated with a significant reduction of fluorescence lifetime.

5. For Figs. 6 and Suppl. Fig. 9, the color scales is best kept constant and chosen to maximize color contrast – it’s difficult to see cyan on a background of green. (minor)

The color scale for FLIM images was kept constant for AF488 and Atto647N and already maximized for the contrast. We cannot further improve the contrast, but believe the blue (short lifetimes) pixels can be well distinguished from the green (long lifetimes) pixels in the images shown in Fig. 6 and new Supp. Fig.8. In addition, it is worth keeping in mind that readers with protanopia or deuteranopia, the most common forms of color blindness, are able to distinguish blue, cyan and green.

6. For the FRET data presented in Suppl. Fig. 10, because of the high initial concentrations of labeled proteins (i.e., 1 micromolar each of AF488-labeled alpha-synuclein and Atto647N-labeled Tau variant), which translates to even higher concentrations in the condensates, changes in donor lifetimes can not only be due to intermolecular FRET but also, if not dominated by, fluorescence quenching. Given this, can the observed changes in calculated FRET efficiencies be interpreted as being mainly due to changes in protein conformation?

We have not interpreted the changes in FRET efficiency as changes in protein conformation, but rather an indication of changes in the protein network between droplets and rafts and puncta. We agree with the reviewer in that the decrease in donor lifetime is likely related to an increase in fluorophore density in the puncta. Indeed, this has been our rationale to explain the decrease in fluorescence lifetimes for both donor and acceptor in the puncta as compared to that of the droplets and rafts (see comment #4). For additional clarity, in the revised manuscript, we have modified a previous sentence and added a new sentence to more explicitly relate the decreased donor lifetime to a higher concentration of fluorophore in the puncta.

“Interestingly, this change was dramatically enhanced for the puncta inside the rafts (Fig. 6c), suggesting that further protein condensation occurs in the puncta.”

“The reduced fluorescence lifetime in the puncta was concomitant with increased intensity, particularly for Atto647N-labeled Tau441 (Supp. Fig. 8a), and higher FRET efficiency values, for both α S/Tau441 and α S/ Δ N_T-Tau systems, suggesting a further condensation of the proteins in the interior of the electrostatic coacervates already at 5 h after LLPS was triggered.”

Reviewer #2 (Remarks to the Author):

In their manuscript titled 'Molecular Mechanism for the Synchronized Electrostatic Coacervation and Amyloid Co-Aggregation of Alpha-Synuclein and Tau, Gracia et al characterize the co-condensation of recombinant alpha-Synuclein and Tau in vitro. They use a set of sophisticated biophysical methods to describe the protein conformations of aSyn and Tau in condensates. They find that negatively charged aSyn coacervates with Tau in crowding conditions, that the produced condensates retain liquidity for a long time, and that over time Tau aggregates of Tau form in solutions with aSyn, Tau, and PEG. Although the techniques used are of interest for the LLPS community, the findings bare little new information: it is known that both Tau and aSyn independently and together form condensates that transition into aggregates, and it has been shown that Tau coacervates with a number of polyanionic polymers, such as aSyn. In summary, the data seem incomplete, provide not enough novelty, and do not support the conclusion.

At the time we submitted this manuscript, our manuscript was reporting that α S is able to form protein droplets by complex electrostatic coacervation with positively-charged IDPs for the first time. During the reviewing process, a paper with this finding was published in *Nature Communications* (<https://doi.org/10.1038/s41467-022-28797-5>). In this paper, the complex coacervation of α S and the prion protein was demonstrated. These findings, in part similar to what we have found for α S and Tau (although this last interaction has more disease implications), were published in the same journal we have our manuscript in review, highlighting

the novelty of the findings and the suitability of the journal to publishing our results. In addition, we are reporting the nucleation of α S/Tau hetero-aggregates inside the liquid coacervates for the first time, and have identified some of the key properties that control the aging of the α S complex coacervates. Depending on the complexity of the protein network and affinity of the protein interactions, the initially liquid droplets can be kinetically arrested in metastable gel-like condensates, where aggregation is hindered, or suffer fast coalescence, generating large protein reservoirs that maintain liquid properties allowing for the re-arrangement of proteins and aggregation. We believe the results and findings we are reporting in this manuscript are sufficiently novel and relevant for publication in this journal.

Major concerns:

The most problematic major confound in the study I see is the extremely high % of PEG that the authors use to induce Tau-aSyn co-condensation. Cellular crowding creates a colloid pressure comparable to 1-2% PEG, whereas the authors use 15% PEG. In these conditions, Tau will co-condensates with almost every negatively charged polymer.

While we have performed most of the experiments at 15 % PEG, similar results, although with a reduced fraction of condensed phase in the solution, are observed with lower crowder concentrations. We have now included a panel in Supp. Fig. 2 (panel c in the new version) showing the formation of condensates at 5 % PEG and their resistance to the treatment with 1,6-hexanediol.

The well-established estimation of macromolecule concentrations, and the corresponding excluded volume effects, for the cytoplasm of cells is ca. 0.3 g/ml (see for example Zimmerman S.B. and Trach S.O. *J. Mol. Biol.* 1991, 222, 599-620; Pielak GJ et al., *Biochemistry* 2009, 48, 226-34; Luby-Phelps K et al., *Int. Rev. Cytol.* 2000, 192, 189-221). 15 % PEG solutions correspond to 0.15 g/ml, well within the estimated macromolecular crowding effect inside cells, and we have reported that similar LLPS effects are observed at lower concentrations (for example 5 % PEG, corresponding to 0.05 g/ml). Importantly, we have excluded any possible associative or segregative effect of the crowders in the LLPS process of α S and Tau/pLK, and observed similar effects for two structurally different crowder molecules (PEG and dextran), which suggests that the main contribution of the crowders to facilitate LLPS in our systems is related to their excluded volume effects.

Another major confound is that there are no data presented on aSyn and Tau alone, especially for the liquid-to-solid-phase-transition experiments. Reading other literature on Tau condensation, it seems that aSyn in fact may inhibit aggregation of Tau rather than promoting it, as suggested by the title.

α S alone is not able to trigger LLPS at the low protein concentrations used in this study, and although Tau alone is capable, we have demonstrated that it can also do it with α S, being α S a scaffolding protein in the coacervates. We showed that Tau alone can phase-separate (Fig. 2c, left panel, Supp. Fig. 2b) and form aggregates inside the coacervates (previous Supp. Fig. 8e, first row, now Supp. Fig. 9 in the revised version of the manuscript) already in the previous version of the manuscript. Importantly, we were reporting that α S can phase-separate in the presence

of other disordered cationic polypeptides such as pLK and focused the paper on this feature. This was a novel finding at the time we submitted the manuscript. While in the process of reviewing the manuscript, a research article showing that α S can phase-separate with a cationic disordered polypeptide, concretely the prion protein, was published in *Nature Communications* (Agarwal A. et al. *Nat. Commun.* 2022, 13, 1154), showing the relevance and novelty of our independent findings. In our manuscript we generalize the ability of α S to trigger LLPS together with a cationic disordered polypeptide by electrostatic coacervation and we show the relevance of the nature of the protein network (multivalence, charge screening and affinity) of the condensed phase to govern the rate of liquid-to-solid phase transition, thus generating either gel-like condensates or long-live liquid condensates where amyloidogenic proteins can reorganize and trigger the formation of amyloid-like aggregates inside the liquid coacervates. We showed in the previous version of the manuscript that Tau homotypic coacervates can result in Tau aggregation inside the coacervates (see Supp. Fig. 8). But we report for the first time that hetero-aggregates composed of α S and Tau can be formed inside the liquid α S/Tau condensates; a finding that could have relevance in some neurodegenerative diseases.

In the revised version of the manuscript, we have included new data to further probe the amyloid-like nature of these aggregates and characterize the composition of these aggregates by single-molecule fluorescence techniques, which has also allowed us to estimate the fraction of hetero-aggregates and homo-aggregates, composed of only Tau or α S, formed inside the α S/Tau coacervates (see new Fig. 6 and new Supp. Fig. 10 and 11).

Furthermore, no evidence of 'amyloid co-aggregation' is given, but it seems that amorphous protein aggregates of Tau forming in the highly crowded solution in fact exclude aSyn (see FLIM images).

In the previous version of the manuscript, we showed that the punctual aggregates formed inside the liquid coacervates were able to bind ThT, as the amyloid aggregates do (Figure 5). In order to further investigate this idea, we isolated these aggregates from the liquid coacervates and obtained a dispersed solution of aggregates (previous Supp Fig 8). With these samples of isolated aggregates, we tested the ability of these aggregates to bind ThT and increase its fluorescence quantum yield in the typical in vitro ThT experiments. These are well-established assays to check the amyloid nature of dispersed aggregates in a solution. We found that the LLPS-derived isolated aggregates were able to bind and increase the fluorescence quantum yield of ThT like canonical amyloid fibrils do. Also we showed that these aggregates are resistant to high salt concentrations, indicating that non-electrostatic forces are contributing to the stability of these aggregates, but sensitive to high concentrations of chemical denaturants, similar to the amyloid-like aggregates.

We have tried to perform other typical analyses for testing the amyloid structure of the aggregates, but the concentrations of the isolated aggregates in the samples are, unfortunately, very low (in the low nM range). Attempts to concentrate them by centrifugation or filter devices have failed. The excess of residual monomeric protein in the solution also hampers average structural characterizations. We have, however, successfully performed new single-molecule fluorescence experiments of the isolated aggregates and estimated the fraction of hetero-aggregates composed of α S and Tau. Most of the aggregates generated inside the coacervates were hetero-aggregates composed of both α S and Tau (ca. 60%). Also, ca. 10% of the aggregates contained only α S, and ca. 30% only Tau. These experiments have provided relevant information

on the composition of these aggregates and we have included them in the revised version of the manuscript (see new Fig. 6); see also Figure 1 in this document.

In order to corroborate further the amyloid nature of the aggregates generated in the interior of the α S/Tau441 liquid coacervates, we have performed new experiments with a well-established Tau variant that has been demonstrated to be unable to form amyloid aggregates. In this variant, the main regions responsible for Tau amyloid aggregation, residues 275-311, have been removed (von Bergen M et al., *BBA* 2005, 1739, 158-166; Mukrasch M et al., *JBC* 2005, 280, 24978-86; Zibae S et al., *Prot. Sci.* 2007, 16, 906-18; Seidler P.M. et al., *Nat. Chemistry* 2018, 10, 170-176). We have generated this variant, that we have referred to as Aggregation Deficient Tau variant, or AggDef-Tau, and reproduced previous results on the inability of this variant to initiate amyloid self-assembly in the typical in vitro experiments for Tau amyloid formation (new Supp. Fig. 11a-b). We have also observed that the LLPS properties of Tau are not altered in this variant, as expected since the deleted region has not major contributions in this process, and, thus, it forms liquid droplets and rafts with α S with identical properties to those of the full-length Tau protein (new Supp. Fig. 11c-d). However, we could not detect the formation of puncta by FLIM and when performing the aggregate isolation protocol used for isolating α S/Tau441 puncta (aggregates) from the liquid coacervates, only few aggregates were recovered (almost 10-times less than with Tau441), which also mainly contained α S (new Supp. Fig. 11e), in contrast to the α S/Tau441 aggregates that were rich in Tau (new Fig. 6). These results indicate that the amyloid-essential region of Tau protein is also essential for the formation of α S/Tau aggregates inside the liquid coacervates, further demonstrating the amyloid nature of these aggregates. These results can now be seen in Supp. Fig. 11 of the revised version of the manuscript; see also Figure 2 in this document.

Also, Thioflavine fluorescence intensity increases upon inhibition of its free rotation in solution, and therefor usually also indicates an increase in viscosity in condensates, and it does not only label amyloid structures. Dot like Thioflavine+ structures cannot be taken as a sign of amyloid aggregation (they may resemble any aggregation or sticking of the Thioflavine) if no comparison to conditions with lower Thioflavine signal is shown for comparison. Especially for the distinction of liquid condensates and aggregates, the use of Thioflavine fluorescence is highly controversial since both conditions largely increase Thioflavine fluorescence.

We agree with the reviewer that the use of ThT in LLPS solutions is not appropriate to indicate whether the proteins suffer amyloid aggregation inside the condensates when an increase in the fluorescence quantum yield of the ThT is observed in the condensates. As pointed out by the reviewer, ThT is a fluorescence molecular rotor, whose ability to emit fluorescence is inhibited by internal quenching due to the free rotation of an internal bond in the fluorophore. Upon restrictions in the rotation of this bond, either by binding to amyloid aggregates or by an increase in solution viscosity, the fluorescence quantum yield of ThT increases. As the viscosity of the coacervates is typically higher than in the diluted phase, it is not surprising that coacervates could show an increase in ThT signal without suffering processes of amyloid aggregation. Given this problematic about the use of ThT in LLPS solutions, we decided to probe the ability of the aggregates generated inside the α S/Tau coacervates to bind ThT after isolating these aggregates from the coacervates and obtaining a dispersed solution of aggregates under non-LLPS conditions (previous Supp. Fig.8, see Supp. Fig. 10 of the revised version of the manuscript).

The aggregates were extracted from the coacervates by the treatment of the LLPS solution with 1M NaCl. Under such conditions, the electrostatic coacervates dissolve but the aggregates remain stable, indicative that the forces that stabilize the aggregates are non-electrostatic in nature, in contrast to the forces that govern LLPS. In this way, we obtained a dispersed solution of aggregates, being these aggregates those that were generated inside the α S/Tau coacervates. In these dispersed aggregate solutions, we added ThT and analysed the ability of the fluorescent probe to bind to the aggregates and determined if the binding yields to an increase in fluorescence quantum yield similar to that of the typical amyloid fibrils, indicative that the dye binds similarly (similar restrictions in the movement of the chemical bond of the dye responsible for the internal quenching) to both types of aggregates. This was the case, as we showed in the previous version of the manuscript (previous Supp. Fig. 8, see Supp. Fig. 10 of the revised version of the manuscript). Note that the experiments performed are typical *in vitro* ThT experiments for testing the amyloid nature of aggregates in a dispersed aqueous solution.

We understand the concerns raised by the reviewers about the amyloid nature of the aggregates from the experimental evidences showed in the previous version of the manuscript. We believe that we have now provided additional evidences that indicate that the aggregates generated inside the α S/Tau441 condensates have an amyloid-like nature and made an effort to clarify previous evidences that seemed to not be clear enough through the text. We have also considered removing the word “amyloid” from the title, although we believe that we have enough experimental evidences to describe them as amyloid-like aggregates throughout the text.

Reviewer #3 (Remarks to the Author):

The article of Gracia P. et al entitled “Molecular mechanism for the synchronized electrostatic coacervation and amyloid co-aggregation of alpha-synuclein and Tau” addresses a very interesting topic in the context of understanding the formation of amyloid aggregation from protein condensates. The paper is very well written in a clear progressive way that allows the reader to understand the different steps containing a lot of data (and supplementary data) from different biophysical techniques. The results are clearly presented and discussed and all the figures are very clear and convincing. The work is focused on the behavior of alpha-synuclein (alphaS) when it phase separates into liquid condensates with different positively charged polypeptide: poly-L-lysine and the protein Tau either full length or truncated from its N-ter part. They demonstrated the electrostatic nature of the coacervates and that alphaS plays a scaffolding role. They observed the formation of amyloid co-aggregation (alphaS/Tau) inside the liquid coacervates and determined the key factors leading to the formation of the aggregates. All the results brought by this study led to the proposition of a model for the synchronized formation of either liquid-like or gel-like structures. This mechanism has a potential relevance in the context neurodegenerative diseases involving amyloid formation of alphaS. As my field of expertise is EPR spectroscopy and in particular SDSL-EPR, my main questions are on this part of the article. The results obtained from the labeling of two Cys variants of alphaS showed very little or no spectral modification resulting from the phase separation leading to the information that alphaS keeps a high degree of flexibility.

1. Could the authors justify the choice of their nitroxide labels ? Why did they choose TEMPOL rather than the most currently used MSTL ? Maybe spectral variations could be better detected using MTSL thanks to its higher flexibility compared to maleimido-TEMPOL.

The main reason that led us to choose the probe maleimido-TEMPOL for EPR experiments is that protein Tau is purified and stored in a buffer containing DTT to maintain its two natural cysteine residues in a reduced state and prevent the formation of disulfide bonds between protein molecules. We decided for maleimido-TEMPOL since the thioester bond to the cysteine is more stable than the disulfide bond of MTSSL in the presence of DTT.

To make this point clear, we have included the following text in the EPR section of the Materials and Methods:

“Single cysteine α S variants were spin-labeled with 4-hydroxy-2,2,6,6-tetramethylpiperidine-N-oxyl (TEMPOL) at positions 24 (TEMPOL-24- α S) and 122, respectively (TEMPOL-122- α S). This spin label was chosen due to its more stable bond in the presence of DTT, which is present in the storage buffer of Tau in these experiments.”

2. The authors excluded the effect of viscosity on the (slight) spectral shape modification at position 122. I was thinking on another potential effect of broadening that could come from spin-spin interaction between labeled proteins in the droplets characterized by a high protein concentration. Could the authors reject this possibility ?

This is, indeed, a very interesting question. In the case of LLPS of α S with pLK, we attribute the observed broadening at position 122 entirely to a slowdown of the dynamics brought about by the interaction of pLK with the C-terminal tail of α S. We can say spin-spin interaction does not contribute significantly in this case because such broadening is not observed when the label is placed on residue 24 of α S, where the average overall spin-spin distance would be similar. On the other hand, in the case of LLPS of α S with Tau, a contribution to the broadening due to spin-spin interaction cannot be ruled out on the basis of the experiments shown in the manuscript.

To answer the reviewer's question, we have performed new spin-dilution experiments that have shown that, indeed, spin coupling contributes partially to the observed broadening in LLPS with Tau (see figures below). However, the broadening of α S spectra in the dispersed phase (see affinity measurements in Fig. 3e with excess of Tau), where the local spin concentration is not high enough for the spin-spin interaction to play a role, can be entirely attributed to the slowdown of the dynamics of position 122 due to α S/Tau interaction. This makes us think that dynamic changes of α S brought about by the interaction with the different poly-cations are contributing to broadening in both, the condensed and dispersed phase. In LLPS conditions, this is evident from the fact that, for example, the position 122 does not comply with the isotropic model as position 24.

Figure 3. Comparison of EPR spectra of α S labeled at position 24 in different conditions. 100% labeled- α S in 15% PEG (α S in the diluted phase, blue). 100% labeled- α S in 15% PEG in the presence of Tau at 1:1 ratio (most of the α S in the condensed phase, yellow). 25%-labeled α S (75% unlabeled α S) in 15% PEG in the presence of Tau at 1:1 ratio (most of the α S in the condensed phase but only 25% labeled, red).

Figure 4. Comparison of EPR spectra of α S labeled at position 122 in different conditions. 100%-labeled α S in 15% PEG (α S in the diluted phase, blue). 100% labeled- α S in 15% PEG in the presence of Tau at 1:1 ratio (most of the α S in the condensed phase, yellow). 25%-labeled α S (75% unlabeled α S) in 15% PEG in the presence of Tau at 1:1 ratio (most of the α S in the condensed phase but only 25% labeled, red).

These findings regarding spin-spin effects are very interesting and therefore, we are currently working on studying separately the dynamics and spin-spin contributions to the lineshape for α S/Tau condensates. We are performing new experiments and simulations to further elucidate the stoichiometry and more details of α S/Tau interaction and potential α S/ α S interactions that could lead to nucleation at an early stage of the process and eventually to the formation of aggregates. However, this is still work in progress and we consider that the take home message

of the EPR data presented in this article should be that the reduction of rotational spin dynamics contributes to the observed broadening in all our experiments and is primarily due to interactions between proteins rather than to a micro-viscosity change experienced by the probe within the condensate.

To make this point clearer we have changed the paragraph as follows:

“This suggests that there are preferred locations in the configurational space of the spin at the α S C-terminal region in the presence of poly-cations. When taking into account the fraction of α S in the condensed phase under the EPR experimental conditions (89 ± 13 %, 81 ± 6 % and 54 ± 6 % for α S/Tau441, α S/ Δ Nt-Tau and α S/pLK, respectively - see data analysis in Supp. Fig. 2e), it becomes evident that the broadening detected by EPR reflects primarily the interaction of the C-terminal region of α S with the various poly-cations in the condensed phase (major changes when TEMPOL-122- α S is used), rather than an increase in micro-viscosity experienced by the probe upon protein condensation. As expected, when adding 1 M NaCl to the mixture, the EPR spectrum of the protein in non-LLPS conditions is completely recovered (Supp. Fig. 4b). Overall, our data indicates that the changes detected by CW-EPR reflect the interaction of the C-terminal region of α S with the various poly-cations in the condensed phase and that this interaction seems to be stronger with pLK than with Tau.”

3. Line 252-255, the authors concluded that NMR experiments corroborates EPR results. My point is that by NMR they only gained information from the dispersed phase (due to sedimentation problem) whereas EPR results come from both dispersed and condensed phases. This is not clear to me how they can compare results from samples with different contents.

The EPR results from both the dispersed phase under non LLPS conditions (titration experiments in absence of PEG - Fig. 3e), and the condensed phase (>80% of the α S protein is in the condensed phase under LLPS conditions with Tau variants) show spectral changes associated with a reduction of the dynamics due to, at least in part, the electrostatic interaction between α S and Tau/pLK. The same conclusion is obtained from the NMR results, although these results are mostly from the dispersed phase. However, we agree with the reviewer that the comparison between EPR and NMR could be confusing, and we have removed the part of the sentence where we made the comparison from the text in the revised version of the manuscript.

4. In the discussion section (L359), the authors claimed that they worked at low micromolar concentrations and physiological relevant conditions. To my point of view 25-100 μ M protein concentration is still high and I am not sure that it reflects physiological conditions. Could the authors comment what they mean by “physiological relevant conditions” ?

We thank the reviewer for bringing up this important point. The physiological concentration of α S in neuronal synapses has been estimated to be >1 μ M; values as high as approximately 50 μ M have been reported (see: Iwai A. et al., *Neuron* 1995, 14, 467-475; Maroteaux L. et al. *J. Neurosci.* 1988, 8, 2804-15; Withers G. et al., *Dev. Brain Res.* 1997, 99, 87-94; Perni M. et al., *PNAS* 2007, 114, E1009-17.). For example, on the basis of radioactive immuno blots and quantitative mass spectrometry, the intracellular α S concentration was estimated to account for ~0.5% of total brain protein (Iwai A. et al., *Neuron* 1995, 14, 467-475; Schaab C. et al., *Mol. Cell Proteomics* 2012, 11, M111.014068). With a protein concentration of about 100–200 mg/ml

(Zeskind B.J. et al., *Nat. Methods* 2007, 4, 567-9), this corresponds to a physiological α S concentration of 35–70 μ M.

In this study, most of the experiments have been performed at 10 - 25 μ M protein concentrations; within the estimated physiological α S concentration range. The buffers, pH and ionic strength used are also physiologically relevant. We have included a sentence to highlight the physiological concentration of α S in the revised version of the manuscript.

“Here, we present a detailed study of the phase-separating and LSPT behavior of α S in the presence of disordered poly-cations in a controlled environment at low micromolar concentrations and physiologically relevant conditions (note that the estimated physiological concentration of α S is >1 μ M (ref: Iwai A. et al., *Neuron* 1995, 14, 467-475)), following the typical thermodynamically-driven LLPS behavior.

Minor points and typos

Line 476: “degree of labelling was confirmed ...” Could the authors give more quantitative information % of labeling ?

The degree of labeling was estimated by absorbance and MALDI-TOF/TOF to be $>95\%$. We have now included the % of labeling in the revised version of the manuscript.

Line 569 and 570: sub- or upper-scripts are missing for D2O, 1 H and 15 N

Thank-you, this is now corrected in the revised version of the manuscript.

Line 583: “ μ L” instead of “uL”

We are grateful to the reviewer for catching this typo; it is now corrected in the revised version of the manuscript.

Legend fig2: I was confused by the different indications concerning the scale bars of the images. Please check.

We have re-written the sentence in the figure legend concerning the scale bar for improved clarity.

Legend fig3: Band II and band III need to be defined or even indicated in the figure.

We have now indicated the band II and III in the EPR spectrum shown in Fig. 3d.

Sup Fig5 legend. The authors simulated their EPR spectra with a two-component isotropic model. It would be interesting to have the results of the simulation in terms of rotational correlation time and % of population (besides giso and A-values).

For α S labeled at position 24 in the presence of pLK (1:10) in LLPS buffer, the spectra were

simulated with a single component with corr. time $6.5 \cdot 10^{-10}$ s. α S labeled at position 122 in 15% PEG is also simulated with a single component of corr. time $7.5 \cdot 10^{-10}$ s. On the other hand, for 122-labeled α S in 15% PEG + pLK (1:10) the simulation was run with two components: 30% of the previously mentioned component (corr. time $7.5 \cdot 10^{-10}$ s) and 70% of a second, slower, component with a corr. time $2.0 \cdot 10^{-9}$ s.

We have now included this information in the figure legend of the revised version of the manuscript.

Reviewer #4 (Remarks to the Author):

The manuscript by Gracia et al uses a range of experimental approaches to characterize and in part quantify the coacervation between alpha-Synuclein and Tau. The manuscript is overall relatively clearly written and probes a potentially interesting and perhaps (biologically) important interaction. This biological relevance is, however, still somewhat unclear. My major concern with the work is that while many experiments are performed, a number of them are rather qualitative. This makes it difficult to understand what is going on, and the authors use words such as client/driver in a multi-component system w.o. really characterizing the interactions thermodynamically. In addition, in places I find the language overly strong compared to what is actually shown. Overall, the paper probes a potentially interesting interaction which could be important. But in the end, it is somewhat unclear exactly what the authors show. I would recommend that the authors focus their message and tone down claims that are not supported by quantitative measurements and analyses.

In order to obtain more quantitative information on our system, we have performed new single-molecule fluorescence experiments of the isolated aggregates generated in the interior of the liquid α S/Tau condensates, which has allowed us to characterize in detail the fraction of hetero-aggregates composed of both proteins and the fraction of homo-aggregates composed of each of the proteins, as well as the relative composition of the hetero-aggregates. We have also used an additional Tau variant, as explained below.

Major

1. Already in the abstract the authors write “leading to the co-aggregation of both proteins inside the coacervates through a mechanism that could be potentially highly relevant in disease.”. First, the term co-aggregation is somewhat vague and from reading the paper it is still not clear to me whether the authors really show that the two proteins form aggregates “together” (as in with some kind of semi-specific interactions between the two proteins) or whether the experimental conditions simply result in aggregates of both proteins. Also, the statement “that could be potentially highly relevant in disease” is simultaneously vague (“could be potentially”) and strong (“highly relevant”), and I don’t see any evidence from this paper or the discussion that this interaction is relevant for disease. It could well be and obviously both proteins are important. But I think the results do not say much in terms of disease mechanism or progression and would urge the authors to tone down such a link. If not, then at least clarify what they have shown rather than the vague statement at the end of the abstract.

We thank the reviewer for his/her advice, and have modified some sentences, including the final statement in the abstract, to better reflect our findings.

Regarding the characterization of the co-aggregation of both proteins inside the coacervates, as explained above, we have performed new single-molecule fluorescence experiments of the dispersed solutions of aggregates generated inside the α S/Tau coacervates after being isolated from the coacervates by high salt treatment. In the single-molecule fluorescence experiments we are able to estimate the number of AF488- α S and Atto647N-Tau molecules included in each individual aggregate that passes the confocal volume of our microscope during typically 60 min of analysis per sample. Each sample of dispersed aggregates corresponds to the aggregates generated inside the coacervates that were formed in 100 μ L of α S/Tau LLPS samples incubated for 24 h. The dispersed aggregate solutions were then diluted to single-molecule conditions (typically a 1/500 dilution). In each single-molecule analysis, an average of 60 aggregates per sample were detected in the case of α S/Tau coacervates (ca. 10 times less aggregates for the samples that contained the aggregation deficient Tau variant instead of the full-length variant; see response of comment #9). We detected a total of 152 aggregates in three independent experiments of dispersed aggregates isolated from three different α S/Tau LLPS samples. For each aggregate detected, we estimated the number of AF488- α S and Atto647N-Tau molecules that contain the aggregate and the average FRET efficiency of the fluorophores in each aggregate. This last parameter is useful to identify, together with the stoichiometry value between the two types of fluorophores, aggregates that only contain one fluorophore (homo-aggregates) or aggregates that contain both AF488- α S and Atto647N-Tau molecules (hetero-aggregates). We constructed stoichiometry vs FRET efficiency plots to identify the three types of aggregates: only- α S, only- Tau and hetero-aggregates. We found that ca. 60% of the aggregates detected contained both α S and Tau molecules (hetero-aggregates), while 30% contained only Tau and ca. 10% only α S. When analyzing the content of the hetero-aggregates, we observed that the majority of the aggregates were enriched in Tau with respect to α S (see new Fig. 6f and Figure 1 in this document), in agreement with our FLIM data. With these results, included in Figure 6 in the revised version of the manuscript, we are demonstrating that co-aggregates are generated inside the α S/Tau coacervates and that indeed, co-aggregation generating hetero-aggregates is favoured over the homotypic aggregation in the interior of α S/Tau electrostatic coacervates. These new results have been included in the revised version of the manuscript (see new Fig. 6).

2. I think in places that the authors are underplaying the previous results from Zweckstetter et al (ref 36). For example, on p. 2 they write “To the best of our knowledge, the only polymers reported so far to undergo complex coacervation with Tau are RNA molecules or heparin²¹”. While it is true that in Ref 36 aSYN was only shown to partition into condensates formed by Tau and RNA, these authors did show condensates with Tau/RNA/Synuclein. I think it would be important that the authors make it clearer what was previously done.

It was surely not our intention to underplay the previous research article published by Zweckstetter *et al.* and we believe we have given credit for the findings that were reported there. However, we believe that it is important to clarify the role of the different components of a liquid condensate, particularly if a component is part of the scaffolding network of the condensate or a client protein recruited by one of the scaffolding proteins through specific interactions. We have found that α S is indeed a scaffolding protein in the α S/Tau coacervates and we wanted to clarify this finding with respect to what was reported before (note that we

have found that heterotypic interactions between α S and Tau are preferred to the homotypic interactions between Tau molecules in the α S/Tau coacervates, page 5 in the main text of the revised version of the manuscript). We have tried to clarify this point better throughout the text.

With respect to the sentence specified by the reviewer we have modified it as followed:

“In the case of Tau, electrostatic coacervates can be formed through either simple coacervation, where oppositely charged regions of the protein trigger the de-mixing process, or complex coacervation through the interaction with negatively charged polymers such as RNA (21)”.

We have also, modified the following sentence:

“Based on our observation of α S/pLK electrostatic complex coacervation, and the previous observation of α S as a client molecule of Tau/RNA condensates through a direct interaction with Tau, (Siegert, A. et al. *Protein Sci.* (2021) 30, 1326–1336), we hypothesized that α S and Tau could co-separate from the solvent in the absence of RNA by electrostatic complex coacervation, being α S a scaffolding protein in the α S/Tau coacervates (see the Tau charge distribution in Fig. 2e).”

Later (p. 3) the authors write “Owing to this interaction, α S has been recently reported to exhibit a client role and partition into preformed Tau/RNA electrostatic coacervates³⁶”, which is a better description. But in the end, it is about the thermodynamics and interactions and how much the three components and their concentrations shift the phase diagrams, and hence the word “client” is somewhat vague (see also below) and somehow downplays the previous work which I don’t think is fair.

We believe that it is important to distinguish the droplet components according to their role in droplet formation (Banani S.F. et al., *Cell* 2016, 166, 651): 1) scaffolds, which form the backbone of the droplet, and are thus an essential component of the condensate, and 2) client, which is not necessary for the droplet architecture, but recruited to the droplet through affinity with the scaffolds.

We have determined in our study that α S forms the backbone of the droplet (Δ Nt-Tau variant, which is unable to form droplets alone by simple electrostatic coacervation, but it does it with α S by complex electrostatic coacervation), and, therefore, we think it is important to highlight the different role we have identified with respect to what was assigned before. We do not think clarifying this new role means that we are downplaying the previous work, but in any case, we have removed the sentence the reviewer is referring to in this point in the revised version of the manuscript.

In the work by Zweckstetter et al, the authors (I think) obtained condensates of Tau(50 μ M)/polyU(60nM)/aSYN (10 μ M) and 2.5% dextran (and in some experiments 10% dextran). In the current work, phase separation appears to occur at 5–10 μ M Tau, 50 μ M aSYN and 15% PEG (or 20% dextran). Thus, while it is interesting that the current work shows that one can get PS of aSYN/Tau in the absence of polyU, it appears that they instead need to drive PS by addition of greater concentrations of crowding agent. While that is OK (as long as it is made clear), I do think it is important to make it clearer that perhaps the two sets of basic observations

are not so different. Instead, the differences (in my opinion) come from the additional characterization of the condensates here.

Most of the α S/Tau LLPS experiments that we report were carried out with 10 μ M α S and 10 μ M Tau441 (Figure 2, Supp. Fig. 1, Supp. Fig. 2, Supp. Fig. 7) or 25 μ M of each protein (Figure 5 and Figure 6). For FRAP, EPR and NMR experiments, higher concentrations of α S were required for technical reasons. Although most of the experiments were performed at 15 % PEG, lower crowder concentrations can be used to visualize phase separation. We have included in the revised version of the manuscript a panel in Supp. Fig. 2, where we show phase separation in the presence of 5 % PEG. We have performed most of the studies at 15 % PEG because most of the proteins are in the condensed phase under such conditions at low protein concentrations, which has allowed us to perform some key experiments, but the process is also observed at 5 % PEG (and even lower) at the same protein concentrations, although the fraction of the proteins in the condensed phase is, obviously, lower. However, we agree with the reviewer that we had to show the LLPS process at lower crowder concentrations.

3. On p. 3 wording like “co-driving model” is in my opinion vague. As above, it’s likely just about thermodynamics and whether Tau alone, aSYN alone or the two together can phase separate likely “just” depends on the concentrations and the conditions (presence of crowder, poly-U, salt, etc). Unless the authors quantify the thermodynamics via measurements of phase diagrams (beyond the data in Fig. 2c), I don’t think they have evidence to really say what is driving and what isn’t. This is also important on p. 4 where the authors discuss an “active role” vs “client”. Again, such statements would be much clearer to understand if the authors quantified the phase properties/diagrams. I would suggest they either do that or remove claims about what is driving and what isn’t.

Our experimental results indicate that α S and Tau de-mix from the solution forming protein droplets rich in both proteins by a process of electrostatic complex coacervation and that in these droplets both α S and Tau form the backbone of the droplet and are thus scaffolding proteins in the coacervates. The direct evidences for this is clear with the Δ Nt-Tau variant. We have followed the reviewer’s suggestion and standardized the wording throughout the text to better reflect the scaffolding role of α S in the droplets.

Similarly, what do the authors really mean on p. 9 by the statement “Here, we prove that aS is indeed capable of co-driving the LLPS process with Tau through the same type of interactions but with an active role rather than being a simple client molecule.” What do they mean by “co-driving” and “active”?

We were referring to as a co-driving molecule of LLPS or to have an active role in the LLPS process to a molecule that is essential for the LLPS process and forms the backbone of the droplet protein network. Following the reviewer’s previous suggestion, we have standardized the wording throughout the text to better reflect the scaffolding role of α S in the droplets.

4. On p. 3: I really don’t think it is fair to say that 5-15% PEG is a small amount, in particular since most experiments are done with 15% PEG. Indeed, I think that the authors should make it clearer that—at least with the protein concentrations used—they need relatively high concentrations of PEG or dextran to the two proteins to PS. Previous experiments on PEG on IDPs in dilute

solution (e.g. A. Soranno et al, PNAS, 2014) show relatively strong effects. Even better of course, they would quantify the phase behaviour via measuring phase diagrams at higher protein concentrations in the absence or lower amounts of PEG. In the absence of such measurements, they should at least make it clearer how they think the interaction would be physiologically relevant.

We agree with the reviewer in that 5-15% concentrations of PEG are better described as moderate concentrations, and we have removed the words “small amounts” from the sentence the reviewer is referring to.

In the manuscript by A. Soranno et al., PNAS 2014, concentrations of crowders in the range of 5-40 % were used and the conclusion of their study was that a compaction of IDPs is expected in crowding conditions to be similar to that of the cellular interior, as compared to highly diluted conditions. These results, indeed, reinforce the need to mimic macromolecular crowding in *in vitro* experiments for a more physiological understanding of the protein systems.

While the use of macromolecular crowders is widely spread in *in-vitro* experiments to mimic the intracellular crowding effect and it is widely used, therefore, in *in-vitro* studies of protein-driven LLPS, we agree with the reviewer that caution needs to be taken when using macromolecular crowders such as PEG, ficoll or dextran in such experiments to avoid undesirable effects. For this reason, we have carefully analyzed the two crowders used (PEG and dextran) and discarded any segregative or associative effects of the crowders in the LLPS systems (see Supp. Fig 1). Also, we have shown that two structurally different crowders have similar behaviours in our systems, which is consistent with a general common effect related to the steric repulsion phenomenon as the main contributor of the crowders in facilitating LLPS in our study. In the revised version of the manuscript, we have included a panel in Supp. Fig. 2, where we show phase separation of α S and the N-terminally truncated form of Tau in the presence of 5 % PEG with and without 1,6-hexanediol: protein droplets are formed under the two conditions without significant differences.

5. On p.4 the authors argue that the complex coacervation between aSYN and Tau is electrostatically driven. I would like the authors to explain more clearly how they show this. First, addition of salt does not just affect ionic interactions. Second, the Tau condensates alone can also be dissolved via addition of NaCl, so the salt dependency of the aSYN/Tau condensates would likely be affected by salt even if the interactions between aSYN and Tau were not ionic. I realize that the salt dependencies are different, which likely has some evidence for the interaction, but the authors do not really discuss or quantify this. It would be good with a more quantitative argument for the nature of the interaction.

The interaction between α S and Tau has been previously deeply characterized by NMR and proposed to be electrostatically driven (see for example Siegert, A. et al. *Protein Sci.* (2021) 30, 1326–1336) and we here have reproduced the results using NMR techniques (Supp. Fig. 5c-d), as well as EPR (Supp. Fig. 5a-b and Fig. 3e). The interaction is clearly electrostatically-driven as it involves the highly negatively-charged C-terminus of α S and the highly-positively charged central region of Tau. We have included the net charge per residue diagram of each protein sequence (α S in Figure 1a and Tau in Figure 2e) in the revised version of the manuscript to clearly show the charge distribution of the proteins. We have shown that this interaction is the responsible for α S/Tau LLPS. First, we showed that increasing the concentration of salt in the 100-500 mM concentration regime, there is a dramatic reduction of α S/Tau condensation into

liquid droplets, while addition of 1,6-hexanediol at the typical concentrations to disrupt hydrophobic protein interactions does not influence α S/Tau LLPS (Supp. Fig. 2a-c, Fig. 2c). The analysis of the type of interactions that govern protein-driven LLPS typically involves the type of analysis we have used here, since the most common LLPS interactions are either electrostatic or hydrophobic (see for example Dignon G. et al., *Annu. Rev. Phys. Chem.* 2020, 71, 53-75).

In addition, in order to directly prove the electrostatic nature of the interaction between the two proteins that drives LLPS, we used a Tau variant where the negatively-charged N-terminal region was removed (Fig. 2e), since Tau alone can phase-separate by simple electrostatic coacervation (i.e. by electrostatic interactions between the negatively-charged N-terminal region of Tau and the positively-charged central region – this has been proven in previous reports and we have reproduced it). This variant, Δ Nt-Tau, is unable to phase-separate in the absence of a negatively-charge polymer (Fig. 2f, Supp. Fig. 2d). In the presence of α S, the LLPS process is fully recovered (Fig. 2f, Supp. Fig. 2d). However, no LLPS is observed for the C-terminal truncated variant of α S (Fig. 2f, Supp. Fig. 2d). Indeed, the EPR data showed that the interaction between Δ Nt-Tau and α S is lost in the presence of 1M NaCl (Supp. Fig. 4b). In addition, we used an additional Tau variant that corresponds to the amyloid-forming paired helical filament (PHF) region located within the central region of the protein, typically referred to as K18 variant (Fig. 2e). We showed that K18 alone is unable to phase-separate (Fig. 2f, Supp. Fig. 2d), as in the presence of the C-terminally truncated α S, but recovers this property in the presence of full-length α S (Fig. 2f, Supp. Fig. 2d).

An additional evidence is that in the presence of 1 M NaCl, the α S/Tau liquid droplets and rafts are fully-dissolved, but the α S/Tau aggregates remain stable, indicative that the forces that stabilized the aggregates are non-electrostatic and thus inherently different from those that form the liquid coacervates.

We strongly believe that we have enough lines of experimental evidences to demonstrate that the forces that driven α S/Tau LLPS are electrostatic in nature and thus the process corresponds to a complex electrostatic coacervation.

6. On p. 6 a change in translational diffusion of 2x is stated to be “almost identical” and a change in 13x is listed as “slightly faster”. Obviously, these differences are much smaller than the 50–600 fold differences to the dilute phase (these are stated to be “more than 2 orders of magnitude”, but that’s only relative to Tau condensates, not with pLK). I would urge the authors to make the wording somewhat more neutral rather than saying that the 50–600 fold are remarkable (why?) and the 2–13 fold are “almost identical”/“slightly faster”.

We agree with the reviewer on this point, and we have now re-phrased this section as follows:

“As it can be seen from the representative FRAP images (Fig. 3a, α S/Tau441 coacervation) and their corresponding time-course curves (Fig. 3b, Supp. Fig. 3), the dynamics of α S were very similar within the coacervates with Tau441 and Δ Nt-Tau, while significantly faster with pLK. The diffusion coefficient of α S inside the coacervates estimated from the FRAP data (as described by Kang. M and co-workers⁴⁰) was $D = 0.013 \pm 0.009 \mu\text{m}^2/\text{s}$ and $D = 0.026 \pm 0.009 \mu\text{m}^2/\text{s}$ for α S/Tau441 and α S/ Δ Nt-Tau, respectively, and $D = 0.17 \pm 0.04 \mu\text{m}^2/\text{s}$ for α S/pLK systems (Fig. 3c). The diffusion coefficient of α S in the dispersed phase was, however, orders of magnitude faster with

respect to that of all the condensed phases, as determined by fluorescence correlation spectroscopy (FCS, see Supp. Fig. 3) under identical conditions (LLPS buffer) but in the absence of poly-cation ($D = 8.14 \pm 4.33 \mu\text{m}^2/\text{s}$). αS translational dynamics are, thus, greatly reduced within the coacervates as compared to the protein in the dispersed phase, due to a significant molecular crowding effect, although all the coacervates maintain a liquid-like nature within the first half an hour from their formation, presenting αS slightly faster dynamics in the condensates with pLK.”.

7. In the EPR experiments described on p. 6 I lack a description of how much of the protein that is probed is in condensates and how much is in a dilute phase. What do these experiments say about the interactions within the condensates?

The fraction of αS that was in the condensed and diluted phase in the EPR experiments (the paramagnetic probe is located in αS) was reported in the text in page 6: “the fraction of αS in the condensed phase under the EPR experimental conditions ($89 \pm 13 \%$, $81 \pm 6 \%$ and $54 \pm 6 \%$ for $\alpha\text{S}/\text{Tau441}$, $\alpha\text{S}/\Delta\text{Nt-Tau}$ and $\alpha\text{S}/\text{pLK}$, respectively - see data analysis in Supp. Fig. 2d).

Under these conditions, therefore, it is evident that the EPR experiments are able to probe the dynamics of αS in the condensed phase, since at least for the LLPS samples with Tau most of the protein ($> 80\%$) is in the condensed phase. Otherwise, the small fraction of αS in the diluted phase would be too low to obtain a decent EPR signal at the experimental conditions used.

We performed EPR experiments with two variants of αS , one with the paramagnetic probe at the C-terminal region (position 122), where the interaction with Tau and pLK occurs, and the other with the probe at the N-terminal region (position 24), in order to obtain information on the overall dynamics of the protein inside the condensate (not directly influenced by the protein-protein interaction). The EPR results show that while the spectra for the αS variant with the probe at the N-terminal region reflects an isotropic movement, that of the variant at the C-terminal region deviates from this movement as a consequence of the direct interaction with the poly-cations. These results, therefore, corroborate that the C-terminal region of αS is directly involved in the interaction with the poly-cations inside the coacervate (regarding comment #5).

Also, the EPR spectra of the αS variant with the probe at the N-terminal region show small differences with respect to the dynamics of the probe in the absence of poly-cations (no LLPS), which indicates that the local micro-viscosity around the probe in this αS variant has not suffered important changes upon condensation. However, this is not the case of the probe located at the C-terminal region of αS , where the dynamics of the probe are more reduced in the presence of poly-cations. Our data indicate that the spin dynamics changes detected by EPR reflect primarily the interaction of the C-terminal region of αS with the various poly-cations in the condensed phase and that this interaction is stronger (a more significant reduction in the probe dynamics) between αS and pLK than with Tau. Given that our EPR data suggest that the dynamics of the paramagnetic spin at the C-terminal region of αS primarily probes the interaction with the poly-cations, we performed titration experiments to estimate the affinity of the interactions (Fig. 3e).

In the revised version of the manuscript, we have included a final sentence with a summary of the main findings obtained from the EPR experiments:

“Overall, our data indicates that the changes detected by CW-EPR reflect the interaction of the C-terminal region of α S with the various poly-cations in the condensed phase and that this interaction seems to be stronger with pLK than Tau.”

8. On p. 7 the authors write: “When we analyzed the structure and dynamics of the protein that remains in the dispersed phase in the LLPS samples by NMR (Supp. Fig. 5c and d), we observed that the protein behaves almost identically in the presence of pLK and Δ Nt-Tau, both in terms of secondary structure and protein backbone dynamics, as detected by secondary chemical shifts and R1rho relaxation experiments. The NMR data corroborate the EPR results, showing that the C-terminus of α S undergoes the main loss of conformational flexibility, while maintaining the disordered nature as the rest of the protein sequence, under LLPS conditions with poly-cations.”

Given that these experiments report on the protein in the dilute phase, it is unclear in which way they “corroborate the EPR results” (and which EPR results). Also, the sentence “while maintaining the disordered nature as the rest of the protein sequence, under LLPS conditions with poly-cations” it should be clarified that the protein is the same in the dilute phase, and that the experiments do not say anything about the condensates. Similarly, the SI Fig. S5 legend says “LLPS context” which again might be made clearer by saying the dilute phase under conditions where the proteins can phase separate.

The EPR results from both: 1) the dispersed phase under non LLPS conditions (i.e. the titration experiment performed in the absence of PEG, Fig. 3e), and 2) the condensed phase (> 80% of α S is in the condensed phase under LLPS conditions with Tau variants). Under both conditions, the EPR data show spectral changes associated to the electrostatic interaction between α S and Tau/pLK as a stronger reduction in dynamics in the C-terminal region of the protein, as probed by the spin label at position 122, relative to other protein regions, as probed with the label at position 24. The same conclusion is obtained from the NMR results, although these results are indicative of the interaction only in the dispersed phase, although providing information of the interaction at the residue level. However, we agree with the reviewer that the comparison between EPR and NMR could be confusing, and we have removed the part of the sentence where we made the comparison from the text in the revised version of the manuscript.

We have modified the sentences, as suggested by the reviewer, in the main text and in Supp. Fig. 5 legend, as follows:

- “The NMR data shows that the C-terminus of α S undergoes the main loss of conformational flexibility, while maintaining the disordered nature as the rest of the protein sequence, due to its interaction with the poly-cations.”
- “c) Secondary chemical shifts of HSQC spectra and (d) R1p relaxation analysis for 150 μ M C13/N15-labeled α S in the presence of 1.5 mM pLK (green), 75 mM Δ Nt-Tau (light blue) or 225 mM Δ Nt-Tau (dark blue). The data show the reduction of conformational flexibility in the C-terminal region of α S upon interacting with poly-cations, both pLK and Tau.

9. On p. 7/8/9, I would like the authors more clearly to explain how they think about the co-aggregates of aSYN and Tau. They in places seem to imply that there are direct interactions

(FRET; although the values are low) and that they are amyloid (ThT), but it is unclear whether they think they are forming amyloids with both proteins in a single fibril/protofibril or what is going on. How should I interpret the FRET data in Fig. S10 at a molecular level?

We have identified aggregates composed of both Tau and α S that are formed inside α S/Tau coacervates. We first observed that they were ThT stained inside the coacervates (Fig. 5b) and the fluorescence lifetime of the two dyes used to label each protein (in sub-stoichiometric concentrations with respect to unlabeled proteins) was remarkably reduced likely as a consequence of fluorescence quenching due to aggregation (FLIM, Fig. 6a-c). The fluorophores also show a higher FRET efficiency when aggregated (Supp. Fig. 9), demonstrating that the aggregates contain both proteins, whose fluorophores are within the FRET distances for this fluorophore pair (1-10 nm). We also isolated the aggregates from the coacervates and obtained a dispersed solution of aggregates and performed typical in vitro ThT assays. We observed that these aggregates were able to bind ThT and increase its fluorescence quantum yield similarly (indeed virtually identically) as canonical amyloid fibrils do (in Supp. Fig. 10b in the revised version of the manuscript). Also we showed that these aggregates are resistant against high salt concentrations, indicating that non-electrostatic forces are responsible for the stability of these aggregates. However, just like amyloid-like aggregates, they are sensitive to high concentrations of chemical denaturants (in Supp. Fig. 10c in the revised version of the manuscript).

We have tried to perform other typical analysis for testing the amyloid structure of the aggregates, but the concentrations of the isolated aggregates in the samples are, unfortunately, very low (in the low nM range). Attempts to concentrate them by centrifugation or filter devices have failed. The excess of residual monomeric protein in the solution also hampers structural characterizations due to averaging.

To further address this point, we have now performed new single-molecule fluorescence experiments on the isolated aggregates and prove that co-aggregates (hetero-aggregates) composed of α S and Tau are formed inside the α S/Tau coacervates. We also estimated the fraction of hetero-aggregates composed of α S and Tau and found that ca. 60% of the aggregates generated inside the coacervates are indeed hetero-aggregates composed of the two proteins, as compared to ca. 30% of the aggregates that only contained Tau, and ca. 10 % that only contained α S. These results indicate that the heterotypic interactions between α S and Tau are favored in the aggregates over the homotypic interactions. These experiments have provided relevant information on the composition of these aggregates and we have included them in the revised version of the manuscript (see new Fig. 6). We have also included them in this document in Figure 1.

In order to corroborate further the amyloid nature of the aggregates generated in the interior of the α S/Tau441 liquid coacervates, we have performed new experiments with a well-established Tau variant that has been demonstrated to be unable to form amyloid aggregates. In this variant, the main regions responsible for Tau amyloid aggregation, residues 275-311, have been removed (von Bergen M et al., *BBA* 2005, 1739, 158-166; Mukrasch M et al., *JBC* 2005, 280, 24978-86; Zibae S et al., *Prot. Sci.* 2007, 16, 906-18; Seidler P.M. et al., *Nat. Chemistry* 2018, 10, 170-176). We have generated this variant, that we have referred to as Aggregation Deficient Tau variant, or AggDef-Tau, and reproduced previous results on the inability of this variant to initiate amyloid self-assembly in the typical in vitro experiments for Tau amyloid formation (Supp. Fig. 11b). We have also observed that the LLPS properties of Tau are not altered in this variant; this is expected, since the deleted region has no significant contributions in this process, and, thus,

it forms liquid droplets and rafts with α S with identical properties to those of the full-length Tau protein (Supp. Fig. 11c-d). However, we could not detect the formation of puncta (aggregates) by FLIM in LLPS solutions and when performing the aggregate isolation protocol used for isolating α S/Tau441 aggregates from the liquid coacervates and analyzing the dispersed solutions of aggregates by single-molecule fluorescence experiments, only few aggregates were detected in the samples (almost 10-times less than with Tau441), which also mainly contained α S (see new Supp. Fig 11 c,d). These results indicate that the amyloid essential region of Tau protein is relevant for the formation of α S/Tau co-aggregates inside the liquid coacervates, further demonstrating the amyloid nature of these aggregates. In addition, the new results also indicate that, although α S is less prone to aggregate than Tau inside the coacervates, α S aggregates can also be formed.

In the revised version of the manuscript, we have included these new results (see new Fig. 6 and new Supp. Fig. 11).

Given that it is very difficult to provide more experimental evidences to demonstrate the amyloid nature of the aggregates, we have considered to remove the word “amyloid” from the title, although we believe that we have enough experimental evidences to describe them as amyloid-like aggregates throughout the text.

Minor

Abstract: In my opinion, there is no reason to call the methods as advanced biophysical techniques and I would delete “advanced”.

We think that we have used advanced/sophisticated biophysical techniques to characterize LLPS and LSPT in detail: EPR, FLIM, FLIM-FRET, FCS/FCCS. We believe that the new single-molecule fluorescence experiments incorporated in the revised version of the manuscript further support this point.

p. 2: Phase separation does not have to occur through weak interactions.

Although LLPS is generally a consequence of weak, multivalent transient interactions (Guo L. and Shorter J. *Cell*, 2015, 181, 346-61), the word “weak” in the sentence has been removed in the revised version of the manuscript, as suggested by the reviewer.

p. 2: In addition to the examples of FUS and TDP-43, the authors might add hnRNPA1 (and perhaps hnRNPD) as proteins that have been shown to form aggregates and to phase separate, and where there (at least for hnRNPA1) is a link between the two.

HnRNPA1 has now been included as an example of LLPS and LSPT together with FUS and TDP-43.

p. 2: In “One of these proteins is Tau”, it is unclear what “these” refers to. Please clarify.

We have now re-written the sentence to clarify the point raised by the reviewer.

p. 2: In “Tau has been shown to trigger LLPS” it is unclear what is meant by “trigger”. (Same on the use of the word on p. 9)

We have re-written the sentences indicated by the reviewer as suggested.

“Tau has been shown to spontaneously de-mix from the solution/cytoplasm as a consequence of favorable electrostatic interactions...”

“We have found that α S, containing a highly negatively charged C-terminal region at physiological pH, is able to form protein-rich liquid droplets in aqueous solutions by LLPS in the presence of highly cationic disordered polypeptides, such as pLK or Tau, by a process of electrostatic complex coacervation.”

p. 3: The authors use pLK of “100 residues”. How pure is this (i.e. what is the range in degree of polymerization)?

Poly-L-lysine of 100 residues was purchased from Alamanda Polymers Inc, Huntsville, AL, USA. The specified degree of polymerization as determined by NMR from the vendor is 90-110 (see the specification sheet from the vendor: <https://www.alamanda-polymers.com/files/specs/000-kf100.pdf>). We have now included this information in the revised version of the manuscript. We have now included this information in the Materials and Methods section of the revised version of the manuscript.

p. 3: “First, we confirmed the electrostatic interaction of pLK with the Ct-domain of α S by solution NMR spectroscopy (Fig. 1b)”. While I do not doubt that the interaction between pLK and α S has a strong electrostatic component, the NMR experiments do not show it is electrostatic.

We have now re-written the sentence as follows:

“First, we confirmed that pLK interacts with the Ct-domain of α S by solution NMR spectroscopy”

p. 4: “we found that both crowding agents distribute homogeneously over the whole sample without showing a segregative nor an associative behavior”. Can the authors quantify the relative concentrations in dilute and condensed phases to support this?

The quantification by confocal microscopy (fluorescence intensity of the FITC-labeled crowders), showing a homogeneous distribution between the diluted and condensed phase for the two crowders used in this study, was shown in Supp. Fig. 1d, right panels. An analysis of the correlation of the emission intensities between the FITC-labeled crowders (PEG or dextran) and Atto647N-Tau can be seen in the figure below. As it can be seen there is no correlation between the intensities, indicating that the two types of crowders are homogeneously distributed and there is no segregative effects in the LLPS sample (i.e. the crowder does not concentrate either in the diluted or condensed phase, but rather shows equal distribution in both phases).

Figure 5. Correlation analysis between the fluorescence emission intensities of FITC-labeled crowders and Atto647N-Tau in α S/Tau LLPS samples. The analysis corresponds to the figures shown in Supp. Fig. 1d in the manuscript, shown also in the figure in panel a). b) The co-localization analysis was performed with ImageJ to extract the intensity/pixel values in each channel (green-FITC or red-Atto647N) and plotted and analyzed with Excel 2000 9.0.3821 SR-1.

p. 7: “Given such differences in affinities between α S and the various poly-cations, we hypothesized that their liquid properties might evolve differently over time.” Could the authors unpack this argument and how the differences in affinity would lead to differences in time-dependent changes.

We hypothesized that stronger interactions between the poly-ions (as in the case of α S and pLK) could lead to a faster liquid-to-solid transition of the droplets and generate gel-like droplets faster, as we have observed.

p. 7: What is “raft”-like about the “protein rafts”? Why this name? By analogy with lipid rafts?

Yes, we have used the term “rafts” for the pools of coalesced protein coacervates that are generated on the surface of the slide as a consequence of coacervate coalescence and surface wetting. The surface of the slide looks like protein-rich liquid rafts or domains similar to lipid rafts in membranes.

p. 10: How do the authors know the droplets have a low surface energy? And what do they really mean?

We have observed that α S/pLK and α S/Tau droplets are spontaneously and instantaneously generated with liquid-like properties, as they are able to coalesce and wet hydrophilic surfaces. The initial protein network of the droplets is generally not optimized and present regions with uncompensated charges. Indeed, it is reasonable to think that these local electrostatically unstable regions are particularly frequent at the droplet interface (due to the differences in protein densities and then charge densities between the two phases), which results in high electrostatic droplet surface potentials. In order to compensate the charges and minimize the droplet surface potential, droplets can incorporate new polypeptides from the diluted phase, re-organize the protein network to optimize charge-charge interactions, fuse with other droplets or interact with the surface (wetting). We have observed that for all the α S/pLK and α S/Tau systems, the initially formed droplets fuse between each other (see Figure 1 and 2 in the manuscript) and wet hydrophilic, but not hydrophobic, surfaces (Supp. Fig. 7), indicating that the protein network of a number of coacervates is not optimized and the droplets have high electrostatic surface potential. With time, α S/pLK droplets, due to the simpler protein network (only heterotypic interactions between α S and pLK) and the stronger affinity of the protein-protein interactions, seem to be able to balance the charges of the droplets faster (we observed faster α S dynamics within the α S/pLK droplets by FRAP), resulting in relatively small droplets with stabilized protein networks, where the interactions are optimized and thus less transient. The α S/pLK droplets, then, suffer a fast liquid-to-gel transition, as we have observed (see Figure 4 in the manuscript). The α S/Tau droplets, however, are less efficient in optimizing the charge balance of the droplets as a consequence of the higher complexity of the protein network (with both homotypic and heterotypic interactions) and the weaker nature of the interactions. α S/Tau droplets then tend to coalesce and wet the surface in a large extent to decrease their surface potential (see Figure 4 and 6 in the manuscript). The large liquid droplets and rafts that are then generated maintain the liquid nature (see Supp. Figure 7b), as there is a continuous search for the optimization of the charges in the protein network and thus the interactions interchange between adjacent polymers within the network and remain transient.

All the results obtained in our study with the different polypeptide variants (pLK, Tau441, Δ Nt-Tau, Agg-Def-Tau) are in agreement with the proposed model.

This model is in line with theoretical and experimental LLPS studies that have highlighted the relevance of valence exhaustion and electrostatic screening within condensates as a key mechanism to control the size of the condensates and their liquid properties (Ranganathan, S. & Shakhnovich, E. I. Dynamic metastable long-living droplets formed by sticker-spacer proteins. *Elife* **9**, e56159 (2020); Folkmann, A. W., Putnam, A., Lee, C. F. & Seydoux, G. Regulation of biomolecular condensates by interfacial protein clusters. *Science* **373**, 1218–1224 (2021)).

p. 11: On p. 11 what do the authors mean by “coacervate valence”? And in general, what do they really mean by “valence” and “valence exhaustion” in the discussion, and which data reports on this?

Biomolecular LLPS is driven by multivalent biopolymers and several studies have highlighted the “multivalent” nature of the constituent proteins. Valence in this context refers to the multiple associative or “adhesive” domains in the biopolymers that lead to the formation of the droplet network. In the case of α S/pLK and α S/Tau droplets, these domains or valences are constituted by either negatively or positively charged regions. As explained in the previous comment, the fate of the initially formed electrostatic coacervates depends on the rate and degree of

electrostatic balance of the protein network, and this is more generally expressed as valence exhaustion, i.e., the exhaustion of available valences within the protein network or in other words, the exhaustion of valences that are not satisfied through interactions.

We have tried to clarify the discussion on these points in the new version of the manuscript.

p. 14: μM \rightarrow μM , and sub/superscripts missing from D2O, ^1H and ^{15}N

The typos have now been corrected; thanks to the reviewer.

p. S6: In Supporting Fig. 5a, it would perhaps be easier to see what is going on if the data were referenced to (subtracted by) the chemical shift differences in free αS .

Following the reviewer's suggestion, we have referenced the secondary chemical shifts for the interaction of $\alpha\text{S}:\Delta\text{Nt-Tau}$ (1:1.5 molar ratio) to the secondary chemical shifts adopted by αS in aqueous buffer. As it can be seen in the figure below, there are not evident changes, indicating that the conformational propensities in αS are not altered upon binding to Tau.

Figure 5. Secondary chemical shifts of $\alpha\text{S}:\Delta\text{Nt-Tau}$ mixture (1:1.5 ratio) after subtraction of the values for αS in aqueous buffer. No strong changes in secondary structure propensities are evident.

REVIEWERS' COMMENTS

Reviewer #1 (Remarks to the Author):

All my questions/comments/suggestions have been adequately answered/addressed by the authors. I think that the current form of the manuscript has significantly improved from the previous version.

Reviewer #3 (Remarks to the Author):

I thank the authors for their answers to my questions and comments.

I am however very surprised by the answer given to my first question about the choice of the nitroxide spin label (TEMPO rather than most commonly used MSTL). The reason given by the authors is that maleimido-TEMPOL is more stable than the disulfide bond MSTL in the presence of DTT. That is true from the functionalization part of the spin label attached to the Cys residue. But what about the stability of the nitroxide moieties in the presence of DTT ? Usually DTT has to be removed from the protein solution before spin labeling to avoid the rapid reduction of the NO* into the hydroxylamine (EPR-silent) form. Moreover M-TEMPOL is a 6-membered ring nitroxide whereas MTSL is a 5-membered ring's one. It is known that 6-membered ring NO* are even less stable towards reduction than 5-membered ring NO*. The justification given by the authors should be removed (it adds more confusion than before) and I think that the details of the labeling protocols should be given in the material and methods section instead.

My original question was just to discuss about the rigidity of the maleimido function in relation with the spectral shape.

Reviewer #4 (Remarks to the Author):

I thank the authors for their comments. I am still not fully convinced about all the data and interpretations, but believe that the readers should be able to make their own judgements.

As for the new data on the presence of both aSYN and TAU in the aggregates, it might be worth making it clear that the 60% of aggregates that contain both proteins is close to what one would expect from a random preference. This might still be interesting (as amyloid formation is generally specific).

Nevertheless, in the absence of any details on the nature of these aggregates, I am not sure how relevant they are.

REVIEWER'S COMMENTS

Reviewer #1 (Remarks to the Author):

All my questions/comments/suggestions have been adequately answered/addressed by the authors. I think that the current form of the manuscript has significantly improved from the previous version.

We thank the reviewer for helping us to improve the manuscript during the reviewing process.

Reviewer #3 (Remarks to the Author):

I thank the authors for their answers to my questions and comments. I am however very surprised by the answer given to my first question about the choice of the nitroxide spin label (TEMPO rather than most commonly used MSTL). The reason given by the authors is that maleimido-TEMPOL is more stable than the disulfide bond MSTL in the presence of DTT. That is true from the functionalization part of the spin label attached to the Cys residue. But what about the stability of the nitroxide moieties in the presence of DTT? Usually DTT has to be removed from the protein solution before spin labeling to avoid the rapid reduction of the NO* into the hydroxylamine (EPR-silent) form. Moreover M-TEMPOL is a 6-membered ring nitroxide whereas MTSL is a 5-membered ring's one. It is known that 6-membered ring NO* are even less stable towards reduction than 5-membered ring NO*. The justification given by the authors should be removed (it adds more confusion than before) and I think that the details of the labeling protocols should be given in the material and methods section instead.

My original question was just to discuss about the rigidity of the maleimido function in relation with the spectral shape.

The reviewer is correct and we have observed a small but significant reduction in the intensity of the spin signal during the EPR experiments (20 min) due to nitroxide reduction (i.e. a fraction of the spins is converted into an EPR-silent form during spectrum recording) due to the presence of residual DTT in the samples. Every 20 min the signal is reduced ca. 5-6% in our samples. Signal broadening due to spin reduction was not observed, only a systematic decrease in signal intensity with time. After corroborating that only a small intensity decrease was occurring due to spin reduction in our measurements, we decided to maintain residual amounts of DTT in the samples containing Tau441 or Δ Nt-Tau to avoid intermolecular disulfide bond formation in the protein stock solutions containing Tau (with two natural cysteine residues). In all the EPR analysis, spectra intensity was normalized and only signal broadening was considered.

The following sentence has been included in the Methods section to clarify this point:

"Spectra intensity was normalized to avoid differences in spin concentrations across samples and possible spin reductions due to residual concentrations of reducing agents in the samples containing Tau441 or Δ Nt-Tau (present in the stock protein solutions)."

Reviewer #4 (Remarks to the Author):

I thank the authors for their comments. I am still not fully convinced about all the data and interpretations, but believe that the readers should be able to make their own judgements.

As for the new data on the presence of both aSYN and TAU in the aggregates, it might be worth making it clear that the 60% of aggregates that contain both proteins is close to what one would expect from a random preference. This might still be interesting (as amyloid formation is generally specific). Nevertheless, in the absence of any details on the nature of these aggregates, I am not sure how relevant they are.

We thank the reviewer for pointing out this result. Indeed, it is very surprising that most of the aggregates formed inside the liquid aS/Tau complex coacervates are hetero-aggregates instead of the typical homo-aggregates formed in diluted sample conditions. We hypothesize that the heterotypic protein network of the coacervates might lead to the preference for the formation of heterotypic aggregates in their interior. In any case, as we have highlighted in the manuscript, there is a preference for Tau in these amyloid-like aggregates and the results obtained with the amyloid deficient Tau variant indicate that Tau amyloid nucleation is more favourable than aSyn nucleation inside the liquid coacervates.